# Anti-VEGFR2 F(ab')$_2$ drug conjugate promotes renal accumulation and glomerular repair in diabetic nephropathy

Di Liu[1,5], Yanling Song[1,5], Hui Chen[1], Yuchan You [1], Luwen Zhu[1], Jucong Zhang[1], Xinyi Xu[1], Jiahao Hu[1], Xiajie Huang[1], Xiaochuan Wu[1], Xiaoling Xu [2] ✉, Saiping Jiang [3] ✉ & Yongzhong Du [1,4] ✉

Poor renal distribution of antibody-based drugs is the key factor contributing to low treatment efficiency for renal diseases and side effects. Here, we prepare F(ab')$_2$ fragmented vascular endothelial growth factor receptor 2 antibody (anti-VEGFR2 (F(ab')$_2$) to block VEGFR2 overactivation in diabetic nephropathy (DN). We find that the anti-VEGFR2 F(ab')$_2$ has a higher accumulation in DN male mice kidneys than the intact VEGFR2 antibody, and simultaneously preserves the binding ability to VEGFR2. Furthermore, we develop an antibody fragment drug conjugate, anti-VEGFR2 F(ab')$_2$-SS31, comprising the anti-VEGFR2 F(ab')$_2$ fragment linked to the mitochondria-targeted antioxidant peptide SS31. We find that introduction of SS31 potentiates the efficacy of anti-VEGFR2 F(ab')$_2$. These findings provide proof of concept for the premise that antibody fragment drug conjugate improves renal distribution and merits drug validation in renal disease therapy.

Monoclonal antibodies are highly sensitive and specific, applied widely in cancer[1], but their application in kidney diseases is limited owing to the poor renal distribution causing side effects. Although studies in kidney targeting materials have increased thanks to advances in renal anatomy and material science[2], the attempt to reform antibody-based drugs to be suitable for renal disease is limited.

Diabetic nephropathy (DN) is a worldwide renal disease with high morbidity. It is estimated that the number of DN patients will increase by 69% in developed countries and 20% in low-income and middle-income countries by 2030[3]. In the progress of DN, an important characteristic is the overactivation of the vascular endothelial growth factor receptor 2 (VEGFR2) in glomeruli caused by factors including VEGF and Gremlin[4,5], which is associated with endothelial permeability and hence albuminuria[6]. Therefore, VEGFR2 blocking with monoclonal antibodies may be a potential method to relieve DN. However, intact antibody mainly distributed in the liver with little accumulation in the kidney, leading to low bioavailability and side effects. As previously reported, molecular weight is the main factor that influences the renal distribution of drugs or carriers[7,8], and lower molecular weight is related to better renal distribution.

Fragmentation is a simple and effective method to decrease the molecular weight of antibodies. The intact immunoglobin (IgG) antibody (approximately 150 kDa) can be degraded into F(ab')$_2$ fragment (approximately 110 kDa) and several peptides of Fc fragment by pepsin. Furthermore, F(ab')$_2$ retains the epitopes for specific binding and eliminates non-specific binding associated with Fc[9]. Therefore, F(ab')$_2$ may contribute to a higher renal accumulation and a better therapeutic effect for renal diseases.

To the best of our knowledge, the abilities of antibody F(ab')$_2$ fragment in renal targeting and accumulation in DN have not yet been explored. We investigated whether the removal of the Fc fragment contributed to the renal distribution behavior of the VEGFR2 antibody and the F(ab')$_2$ fragment maintained its VEGFR2 binding ability. An antibody tended to be a targeted drug carrier owing to its specificity.

[1]Institute of Pharmaceutics, College of Pharmaceutical Sciences, Zhejiang University, 310058 Hangzhou, China. [2]Shulan International Medical College, Zhejiang Shuren University, 310015 Hangzhou, China. [3]Department of Pharmacy, The First Affiliated Hospital, College of Medicine, Zhejiang University, 310003 Hangzhou, China. [4]Innovation Center of Translational Pharmacy, Jinhua Institute of Zhejiang University, 321299 Jinhua, China. [5]These authors contributed equally: Di Liu, Yanling Song. ✉e-mail: ziyao1988@zju.edu.cn; j5145@zju.edu.cn; duyongzhong@zju.edu.cn

We further prepared anti-VEGFR2 F(ab')$_2$-SS31, an antibody-drug conjugate (ADC) that was developed by conjugating anti-VEGFR2 F(ab')$_2$ with the mitochondria-targeted antioxidant peptide D-Arg-dimethyl-Tyr-Lys-Phe-NH$_2$ (SS31)[10,11] to selectively treat DN. We investigated its efficiency in VEGFR2 blocking, anti-oxidative stress, and DN therapy in vitro and in vivo.

## Results

### Generation and characteristics of anti-VEGFR2 F(ab')$_2$

Anti-VEGFR2, a monoclonal immunoglobulin G (IgG) that can target and block VEGFR2, has the potential to relieve DN. The anti-VEGFR2 used in this study (CD101, Bio X Cell) has been shown to be a potent inhibitor of VEGFR2[12–14]. To investigate the influence of F(ab')$_2$ fragmentation of anti-VEGFR2 in renal distribution ability, we produced the F(ab')$_2$ fragment of VEGFR2 antibody (anti-VEGFR2 F(ab')$_2$) by enzymatic digestion with pepsin. Pepsin cleaved the heavy chains near the hinge region and digested the Fc fragment into small peptides[15]. After ultrafiltration, anti-VEGFR2 F(ab')$_2$ was obtained (Fig. 1a). It was confirmed by sodium dodecyl sulfate polyacrylamide gel electrophoresis (SDS-PAGE) that there was a molecular decrease in anti-VEGFR2 F(ab')$_2$ compared with full anti-VEGFR2 antibody, and by mass spectra (MS) that the light chains peaks were well observed, while heavy chains had weak signals after reduction by Tris(2-carboxyethyl)phosphine hydrochloride (TCEP; Fig. 1a and Supplementary Fig. 1).

### Renal distribution of anti-VEGFR2 F(ab')$_2$

In vivo distribution experiments were carried out to evaluate the renal accumulation efficacy of the full anti-VEGFR2 antibody and F(ab')$_2$ fragment in normal and DN mice, with the rat immunoglobulin (IgG) and its F(ab')$_2$ fragment (IgG F(ab')$_2$) as isotype controls (Supplementary Fig. 2). Before administration, a commercial fluorescent dye of Cy5 was used to label anti-VEGFR2, anti-VEGFR2 F(ab')$_2$, IgG, and IgG F(ab')$_2$, which were abbreviated as anti-VEGFR2-Cy5, anti-VEGFR2 F(ab')$_2$-Cy5, IgG-Cy5, and IgG F(ab')$_2$-Cy5, respectively. The molar concentration and fluorescent intensity of the Cy5-labeled agents were confirmed to be well-matched and uniform (Supplementary Table 1). Fluorescent images of major organs obtained from the normal and DN mice were captured at predetermined times (4 h, 24 h, and 48 h) after intravenous injection of Cy5-labeled agents (Fig. 1b). The healthy mice treated with IgG-Cy5 and anti-VEGFR2-Cy5 had a relatively higher fluorescence intensity in the liver and lung than in other organs after 4 h administration. In contrast, Cy5-labeled F(ab')$_2$ fragments showed much stronger fluorescence intensity in the kidneys at the same time point. In the case of DN mice, the fluorescence intensity of kidneys in IgG-Cy5 and IgG F(ab')$_2$-Cy5 groups had no significant changes compared with normal mice, while in anti-VEGFR2-Cy5 and anti-VEGFR2 F(ab')$_2$-Cy5 groups, the fluorescence intensity of kidneys was increased to some extent compared with normal mice, which might be due to the high expression of VEGFR2 in DN kidneys (Fig. 1c). The fluorescence intensity of anti-VEGFR2 F(ab')$_2$-Cy5-treated kidney was increased much more than anti-VEGFR2-Cy5 group. The improved renal accumulation of F(ab')$_2$ fragmented anti-VEGFR2 in DN may be associated with decreased molecular weight and reduction of the unspecific interaction of the Fc fragment[9]. At 24 h and 48 h post-injection, the fluorescent signals of main organs showed that F(ab')$_2$ fragments retained for a shorter time in healthy or DN mice compared with full antibody or IgG groups, having faster metabolism. The anti-VEGFR2 F(ab')$_2$-Cy5 group had much higher fluorescent intensity in the DN kidney than in other organs, especially at 4 h and 24 h after administration, still showing the best renal accumulation profile compared to other agents.

We further investigated the renal microscopic distribution of anti-VEGFR2 F(ab')$_2$ and the expression of VEGFR2 in DN kidneys by immunofluorescence (Fig. 1d). The results demonstrated the increased VEGFR2 expression in glomeruli of DN mice, which was coincident with western blotting results shown in Fig. 1c. At 4 h after injection, the fluorescent signals of anti-VEGFR2 F(ab')$_2$-Cy5 could be easily identified in glomeruli of DN kidney but barely observed in renal glomeruli of healthy mice. In healthy mice, anti-VEGFR2 F(ab')$_2$-Cy5 was mainly located in vessels, and a similar phenomenon was observed in the kidneys of DN mice treated with anti-VEGFR2-Cy5. In DN mice after administration of anti-VEGFR2 F(ab')$_2$-Cy5 for 24 h, red fluorescent signals could be observed in glomeruli and peritubular capillaries. In addition, the red fluorescent signals of anti-VEGFR2 F(ab')$_2$-Cy5 for 24 were co-localized with the green signals of VEGFR2, indicating the VEGFR2 targeting ability of anti-VEGFR2 F(ab')$_2$.

### Conjugation and characteristics of anti-VEGFR2 F(ab')$_2$-SS31

Anti-VEGFR2 F(ab')$_2$ was conjugated with thioketal modified SS31 to synthesize SS31-conjugated anti-VEGFR2 F(ab')$_2$ (anti-VEGFR2 F(ab')$_2$-SS31) by amide reaction improved according to previously reported method[16,17] (Fig. 2a and Supplementary Fig. 3). We evaluated the anti-VEGFR2 F(ab')$_2$ and anti-VEGFR2 F(ab')$_2$-SS31 by MS after being reduced by TCEP. The abundant ions at m/z 23852.0682 in the spectrum of anti-VEGFR2 F(ab')$_2$ stood for the light chain (Fig. 2b). For the spectrum of anti-VEGFR2 F(ab')$_2$-SS31, the abundant ions at m/z 23879.7249, 25333.1633, and 26030.0983 stood for light chain, double SS31 modified chain, and triple SS31 modified chain, respectively (Fig. 2c). The drug-antibody ratio (DAR) of anti-VEGFR2 F(ab')$_2$-SS31 was calculated to be ~2 according to the MS spectrum. SDS-PAGE showed that the molecular weight of anti-VEGFR2 F(ab')$_2$ was lower compared with the full antibody, while had no significant difference with anti-VEGFR2 F(ab')$_2$-SS31 (Fig. 2d). This may be due to that the molecular weight difference (<3 kD) between anti-VEGFR2 F(ab')$_2$ and anti-VEGFR2 F(ab')$_2$-SS31 was imperceptible in SDS-PAGE.

The binding ability of anti-VEGFR2 F(ab')$_2$-SS31 was confirmed by western blotting assay (Fig. 2e). Mouse VEGFR2 was taken as the loading protein, anti-VEGFR2, anti-VEGFR2 F(ab')$_2$ and anti-VEGFR2 F(ab')$_2$-SS31 as the primary antibody, and HRP labeled anti-F(ab')$_2$ as the secondary antibody. The chemiluminescence signals were well detected in anti-VEGFR2, anti-VEGFR2 F(ab')$_2$, and anti-VEGFR2 F(ab')$_2$-SS31 groups but not in the IgG group, demonstrating that anti-VEGFR2 F(ab')$_2$ and anti-VEGFR2 F(ab')$_2$-SS31 preserved the binding ability of anti-VEGFR2 to VEGFR2.

### Internalization of anti-VEGFR2 F(ab')$_2$-SS31

To evaluate the cellular binding and internalization abilities of anti-VEGFR2 F(ab')$_2$-SS31, we chose mouse renal glomerular endothelial cells (MRGECs) and mouse podocytes (MPC5 cells) as model cells, considering their vulnerability to VEGFR2 overactivation in DN[4,5]. Western blotting results in our study suggested that high glucose considerably upregulated the levels of VEGFR2 in both MRGECs and MPC5 cells (Supplementary Fig. 4). Cy5-labeled anti-VEGFR2 F(ab')$_2$-SS31 was then used to treat normal (VEGFR2-low level) and high glucose-treated (VEGFR2-high level) MRGECs or MPC5 cells, in which anti-VEGFR2-Cy5 and anti-VEGFR2 F(ab')$_2$-Cy5 were used as controls (Fig. 3). Initially, incubation was performed at 4 °C, a temperature that allowed binding but not internalization; anti-VEGFR2 F(ab')$_2$-SS31-Cy5 and anti-VEGFR2-Cy5 and anti-VEGFR2 F(ab')$_2$-Cy5 had a similar profile in binding to high glucose-treated cells, and the fluorescence intensity in high glucose-treated cells was higher than that in normal cells, indicating fragment and chemical modification did not interfere with the binding ability of anti-VEGFR2 to VEGFR2. Internalization of the agents occurred when the temperature rose to 37 °C; incubation under these conditions for 1 h resulted in cytoplasmic localization of the labeled agents in high glucose-treated cells, the fluorescent signals in anti-VEGFR2 F(ab')$_2$-SS31-Cy5 and anti-VEGFR2 F(ab')$_2$-Cy5 group were higher than those in the anti-VEGFR2-Cy5 group, indicating that the

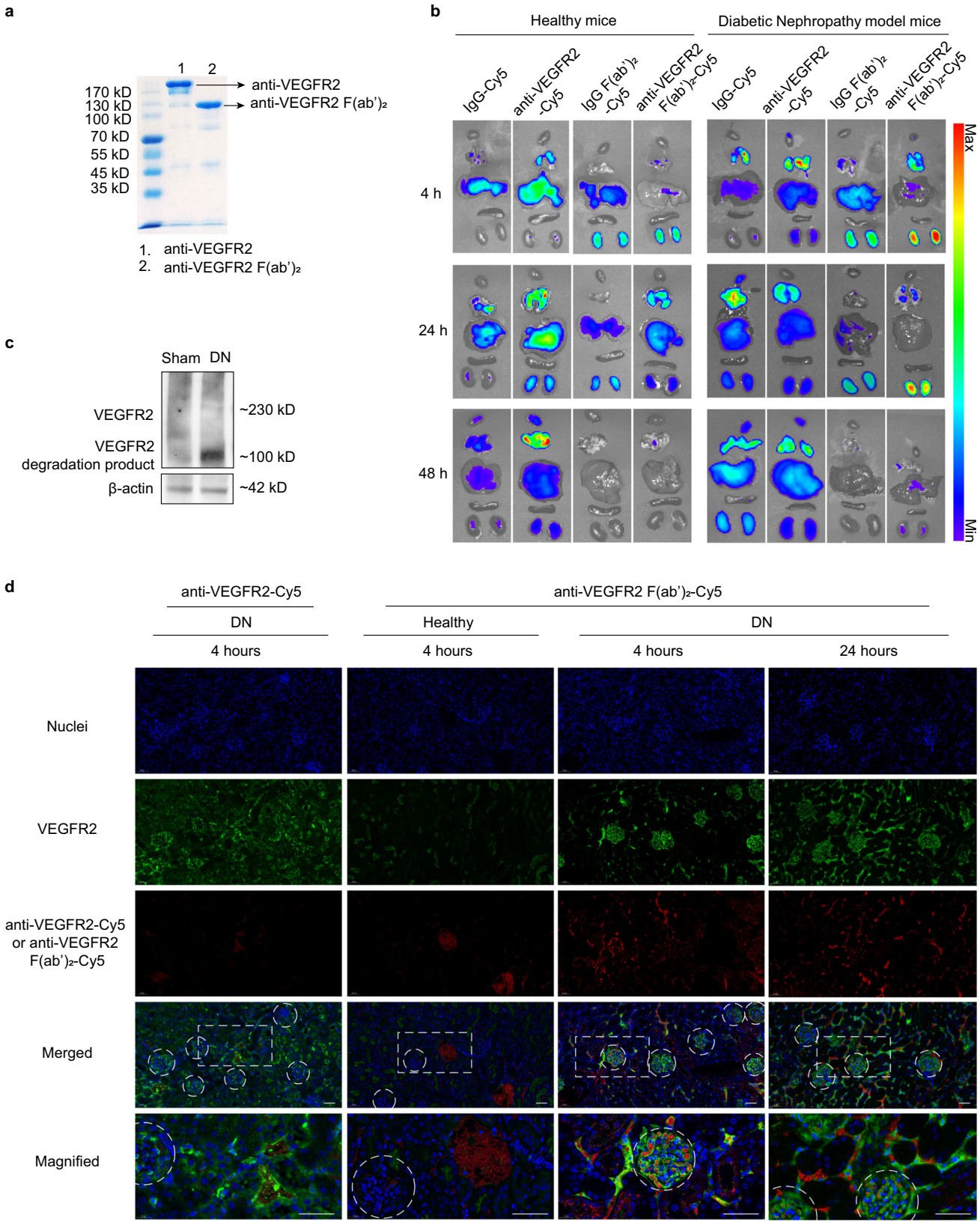

fragment was helpful for cellular internalizing properties of VEGFR2 antibody.

## Cell migration inhibition of anti-VEGFR2 F(ab')₂-SS31

We further investigated the cell migration inhibiting the ability of anti-VEGFR2 F(ab')₂-SS31. In DN glomeruli, VEGFR2 overactivation by VEGF enhanced cell migration, especially endothelial migration, and ultimately induces albuminuria[18,19]. Scratch assay and transwell assay were performed to evaluate the migration capability of MRGECs after different treatments (Fig. 4a–c). It was observed that the migration distance of MRGECs was enhanced by stimulating with VEGF. After treatments with anti-VEGFR2 F(ab')₂ or anti-VEGFR2 F(ab')₂-SS31, there

**Fig. 1 | Renal distribution and glomeruli accumulation of anti-VEGFR2 F(ab')₂.**
**a** SDS-PAGE gel of anti-VEGFR2 and anti-VEGFR2 F(ab')₂. **b** Fluorescent images of the main organs (heart, lung, liver, spleen, and kidney) of mice at 4, 24, or 48 h after intravenous injection of anti-VEGFR2-Cy5, anti-VEGFR2 F(ab')₂-Cy5, IgG-Cy5 and IgG F(ab')₂-Cy5. Streptozotocin (STZ)-induced DN mice model sustained high glucose for 5 weeks, fluorescence-labeled anti-VEGFR2 F(ab')₂ and its control groups were administrated intravenously. 4, 24, or 48 h later, the main organs were harvested for fluorescence visualization. The healthy mice were treated with the same protocol as the control. **c** Western blotting analysis of VEGFR2 expression using homogenates from kidneys of mice subjected to DN and the sham group.
**d** Representative confocal images of kidney sections from mice after intravenous injection of Cy5-labeled anti-VEGFR2 and its F(ab')₂ (red signal) for 4 h. Blue indicates 4′,6-diamidino-2-phenylindole (DAPI) staining. Immunostaining for VEGFR2 is shown in green. White dashed circles denote glomeruli. The bottom images correspond to magnified views of the boxed areas above. Scale bar, 50 μm. The experiments in **a**, **c**, and **d** were repeated independently for three times with similar results. VEGFR2 vascular endothelial growth factor receptor 2, IgG immunoglobulin, DN diabetic nephropathy. Source data are provided as a Source Data file.

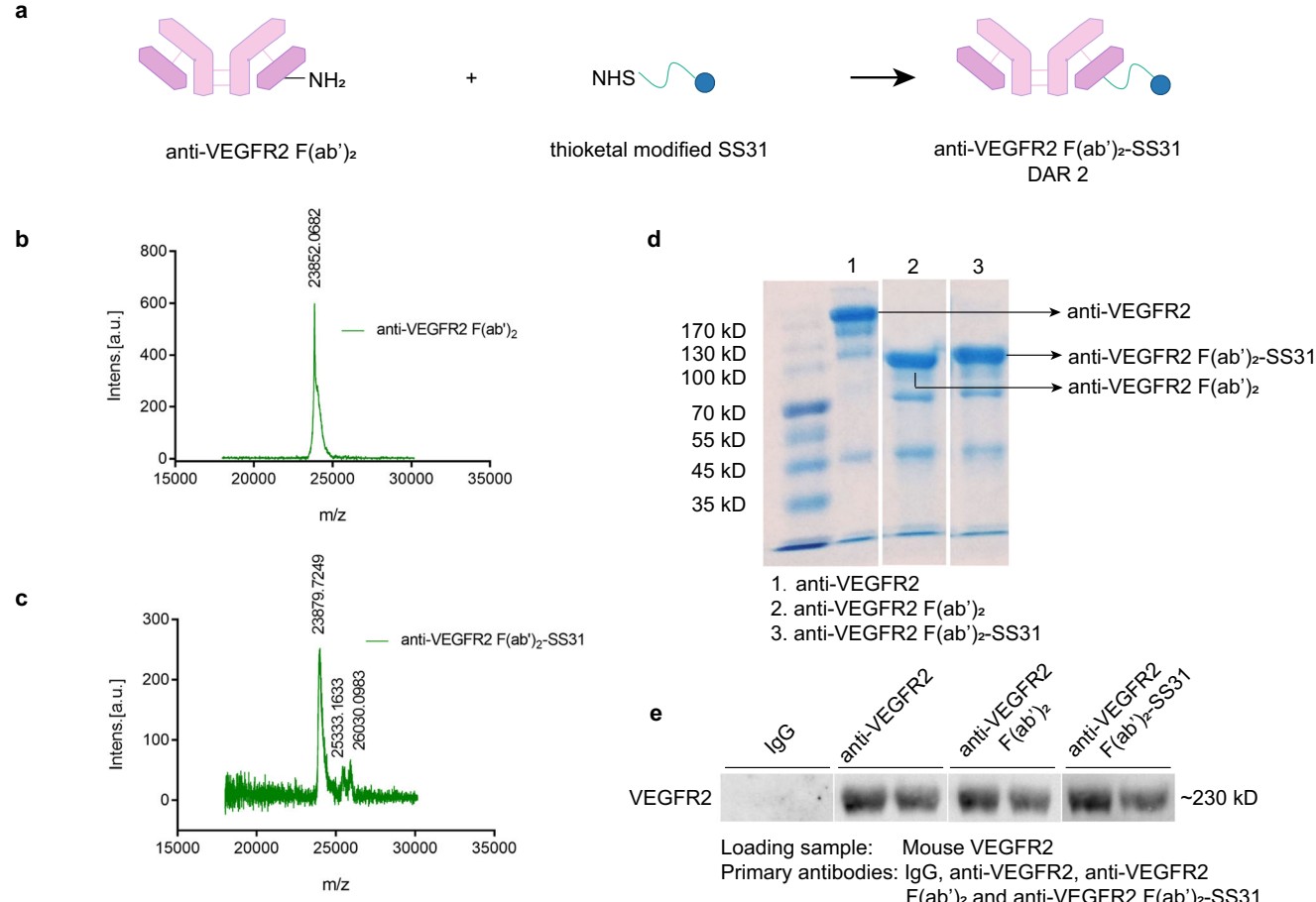

**Fig. 2 | The preparation and characteristics of anti-VEGFR2 F(ab')₂-SS31. a** The construction and synthesis of anti-VEGFR2 F(ab')₂-SS31. Reproduced with permission from ref. 17. Copyright 2018, Oxford University Press. MS data of (**b**) anti-VEGFR2 F(ab')₂ and (**c**) anti-VEGFR2 F(ab')₂-SS31 reduced by TCEP. Detection range from 18 to 30 kD. **d** SDS-PAGE gel of anti-VEGFR2, anti-VEGFR2 F(ab')₂ and anti-VEGFR2 F(ab')₂-SS31. **e** Western blotting analysis of the binding ability of different samples to mouse VEGFR2. The experiments in **d** and **e** were repeated independently for three times with similar results. VEGFR2 vascular endothelial growth factor receptor 2, IgG immunoglobulin. Source data are provided as a Source Data file.

was a dramatic decrease in migration distance, while there were no significant differences in SS31 treated cells, compared with the VEGF group. Transwell-mediated cell migration showed a similar trend with the scratch assay that the migration cells decreased after treatment with anti-VEGFR2 F(ab')₂ or anti-VEGFR2 F(ab')₂-SS31. This observation indicated that anti-VEGFR2 F(ab')₂ and anti-VEGFR2 F(ab')₂-SS31 could inhibit VEGF-induced migration with little influence by chemical modification.

**Anti-oxidative stress and anti-apoptosis of anti-VEGFR2 F(ab')₂-SS31 in vitro**

Oxidative stress has been reported to be a major determinant in the pathophysiology of diabetic nephropathy[20,21]. In this part, the effect of anti-VEGFR2 F(ab')₂-SS31 on in vitro anti-oxidative stress and anti-apoptotic efficiencies was investigated (Fig. 4d–g). Firstly, the ability of anti-VEGFR2 F(ab')₂-SS31, to reduce mitochondrial ROS production in MRGECs or MPC5 cells was investigated by confocal laser scanning microscopy (CLSM) via a mitochondrial ROS probe (MitoSOX). It was observed that the fluorescence intensity of the ROS probe was largely enhanced by incubating the cells in high glucose condition. This result demonstrated that the ROS level in both cells was considerably increased under high glucose condition. The treatment of SS31 or anti-VEGFR2 F(ab')₂-SS31 caused an obvious decrease in the fluorescence intensity, which was lower than that treated with anti-VEGFR2 F(ab')₂. This observation indicated that anti-VEGFR2 F(ab')₂-SS31 and SS31 had a similar ability to reduce mitochondrial ROS production and were better than anti-VEGFR2 F(ab')₂ in MRGECs or MPC5 cells (Fig. 4d, e). Apoptosis has an increased incidence in glomeruli in the kidneys of

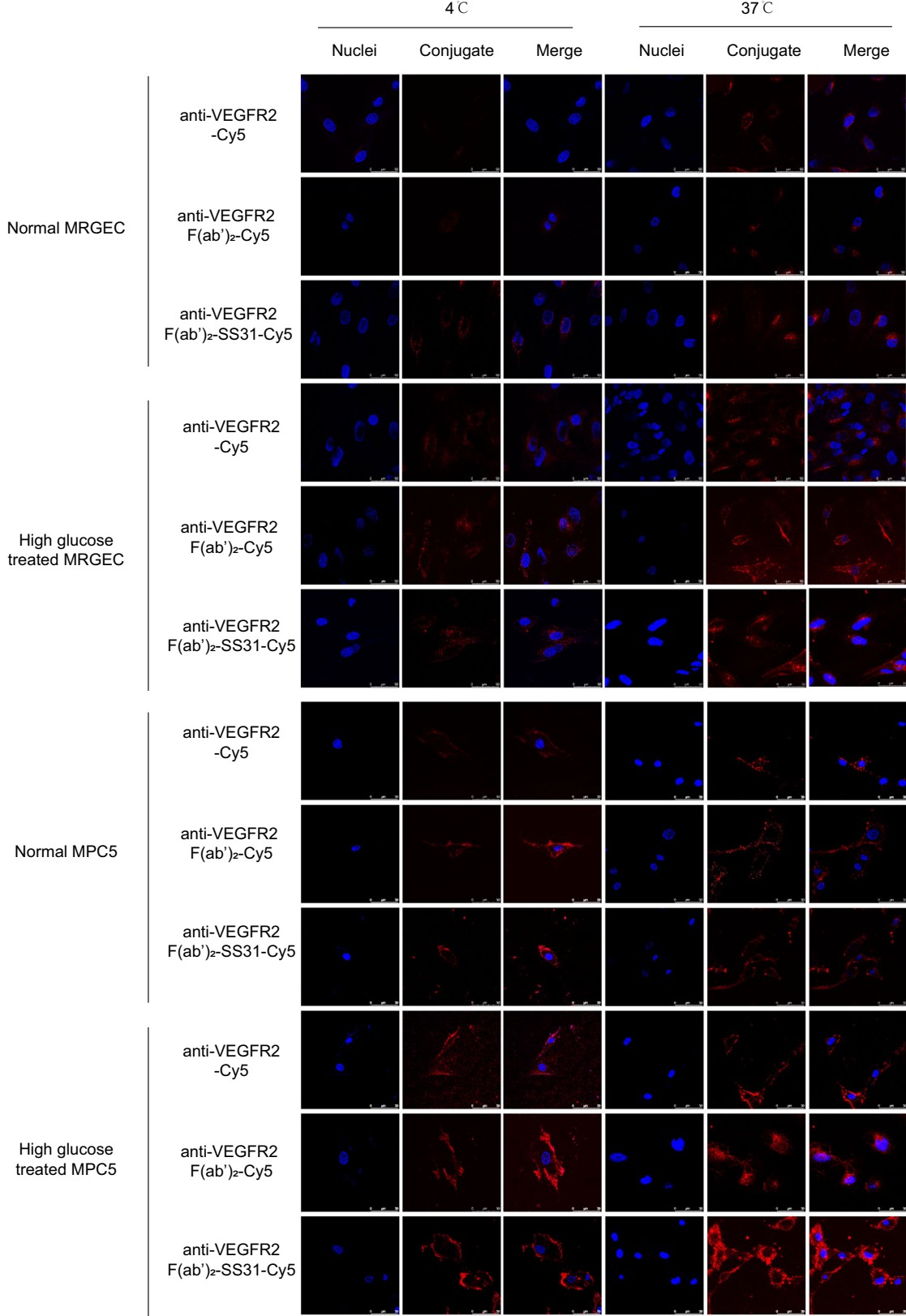

**Fig. 3 | Internalization study of full-length antibody, fragmented antibody, and fragmented antibody-based bioconjugates.** Normal or high glucose-treated MRGECs and MPC5 cells were treated at 4 °C with Cy5-labeled anti-VEGFR2, anti-VEGFR2 F(ab')₂, and anti-VEGFR2 F(ab')₂-SS31 (red); the three constructs bound at this temperature and were found inside the cells once internalization was allowed by incubating at 37 °C for 1 h. High glucose-treated cells showed a better internalization ability than the normal groups. Nuclei, stained with DAPI, are shown in blue. Scale bars, 50 μm. The experiments were repeated independently for three times with similar results. VEGFR2 vascular endothelial growth factor receptor 2, MRGEC mouse renal glomerular endothelial cells, MPC5 mouse podocytes, DAPI 4',6-diamidino-2-phenylindole.

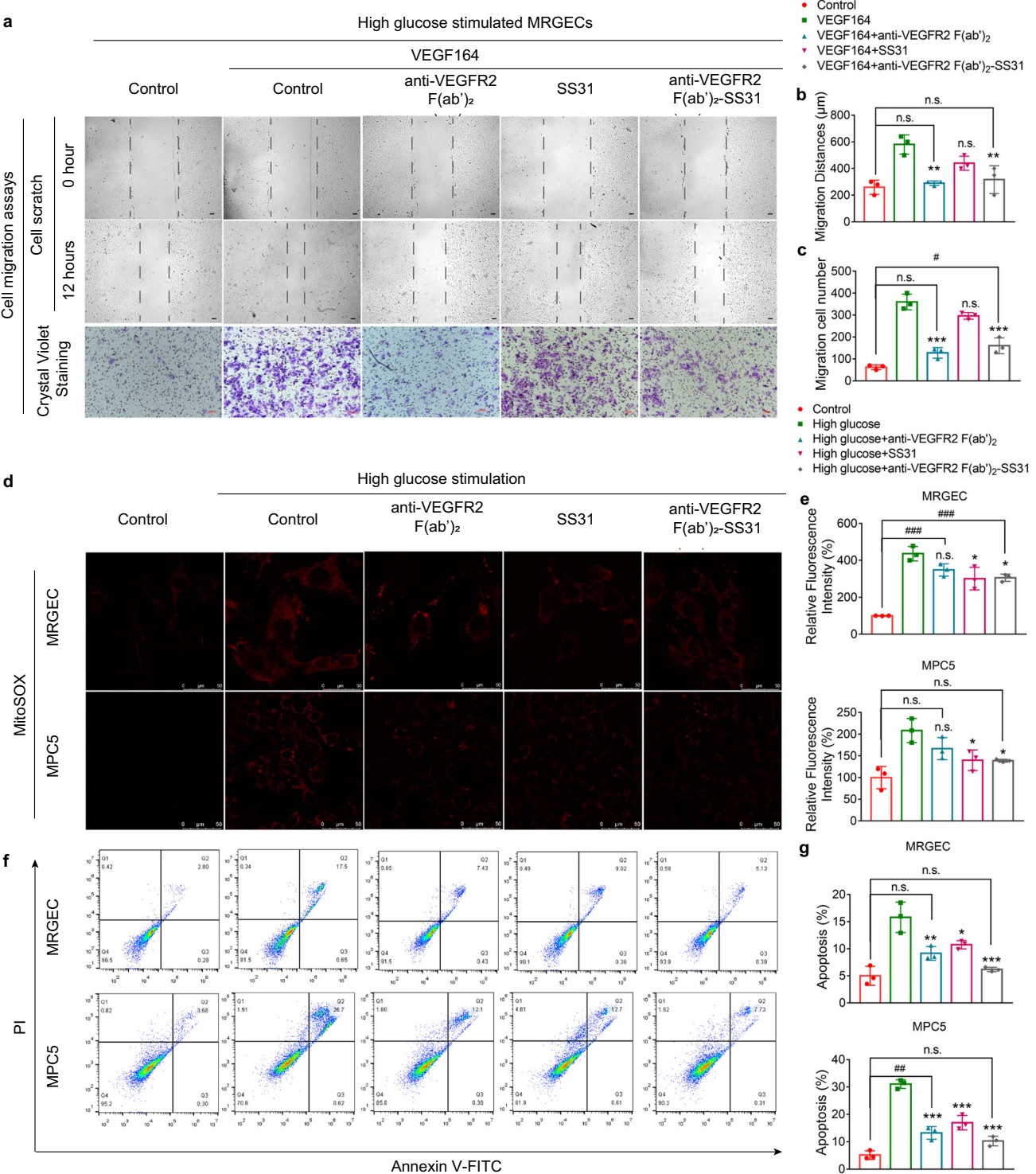

**Fig. 4 | Inhibition of cellular migration and relief of oxidative stress and apoptosis by anti-VEGFR2 F(ab')₂-SS31 in vitro.** High glucose-treated MRGECs or MPC5 cells were treated with anti-VEGFR2 F(ab')₂, SS31, or anti-VEGFR2 F(ab')₂-SS31 at a concentration of 0.4 μM for 12 or 24 h. **a** Light microscope images of the scratch edge in the scratch assay and the migrated MRGECs in transwell migration assay after different treatments. Scale bar, 100 μm. **b, c** Quantitative assay of migration distance of the scratch assay and the migrated cell counts of transwell migration assay in **a. d** Fluorescent images of mitochondrial ROS analysis with MitoSOX in H₂O₂-stimulated cells after different treatments. Scale bar, 20 μm. **e** The MitoSOX mean fluorescent intensity of MRGECs and MPC5 cells in **d. f** Flow cytometry–based

apoptosis assay by the Annexin V-FITC Apoptosis Kit. PI, propidium iodide. **g** The mean percentage of apoptotic cells in **f.** All data are expressed as the mean ± s.d. Statistical significance was calculated using a one-way ANOVA and post-hoc test. $n = 3$ independent experiments in each group, n.s. no significant difference, $*p < 0.05$, $**p < 0.01$, $***p < 0.001$ as compared with VEGF164 or high glucose group; n.s. no significant difference, $\#p < 0.05$, $\#\#p < 0.01$, $\#\#\#p < 0.001$ between groups as indicated. VEGF vascular endothelial growth factor, VEGFR2 vascular endothelial growth factor receptor 2, MRGEC mouse renal glomerular endothelial cells, MPC5 mouse podocytes, DAPI 4',6-diamidino-2-phenylindole. Source data are provided as a Source Data file.

patients with nephropathy[22,23]. Thus, the effect of formulation types on cell apoptosis was examined using flow cytometry. Compared with control groups, high glucose-treated MRGECs or MPC5 cells demonstrated a dramatically increased apoptosis rate. The treatment of anti-VEGFR2 F(ab')₂, SS31, and anti-VEGFR2 F(ab')₂-SS31 significantly decreased the cell apoptosis caused by high glucose stimulation. In comparison, anti-VEGFR2 F(ab')₂-SS31 revealed a greater reduction of cell apoptosis in high glucose-treated cells, indicating that anti-VEGFR2 F(ab')₂-SS31 had the most powerful capability in protecting cell apoptosis from high glucose stimulation (Fig. 4f, g).

### Treatment of DN

The therapeutic effect of anti-VEGFR2 F(ab')₂-SS31 on DN was investigated by using a STZ-induced DN mouse model. After DN modeling, the anti-VEGFR2 F(ab')₂, SS31, and anti-VEGFR2 F(ab')₂-SS31 were administered by intravenous injection once three days for 5 weeks, followed by urine and kidneys collection for further evaluation (Fig. 5a and "Methods" section). All mice after STZ administration displayed a similar profile in the progression of hyperglycemia (Fig. 5b), and weight loss (Fig. 5c), and injecting with different formulations did not significantly alter the glucose profile relative to DN controls, allowing direct comparison of the different mice models.

Albuminuria is the earliest sign of diabetic nephropathy, reflecting damage to the glomerular filtration barrier, and maintaining disease progression through downstream inflammatory and fibrogenic pathways[24,25]. STZ-induced DN mice developed obvious albuminuria compared with the sham group. Anti-VEGFR2 F(ab')₂ and anti-VEGFR2 F(ab')₂-SS31 treated DN mice were protected against albuminuria development with a significantly lower albumin/creatinine ratio compared with nontreated or SS31 administrated DN mice and with no difference compared with sham mice. Histologic examination by light microscopy revealed attenuation of diabetic changes in the kidneys of anti-VEGFR2 F(ab')₂ and anti-VEGFR2 F(ab')₂-SS31 treated mice (Fig. 5d). Morphometric analysis of periodic acid-Schiff stain (PAS)-stained sections demonstrated significant glomerular hypertrophy, hypercellularity, and increased mesangial matrix in nontreated and SS31-treated DN mice, whereas these parameters were significantly reduced in anti-VEGFR2 F(ab')₂ and anti-VEGFR2 F(ab')₂-SS31 DN mice (Fig. 5e, f). Compared with nontreated DN mice, SS31, anti-VEGFR2 F(ab')₂, and anti-VEGFR2 F(ab')₂-SS31 treated mice were partially protected against DN based on improved glomerular morphologic indices of DN, including glomerular basement membrane (GBM) width and foot process width, whereas the parameters were well preserved in anti-VEGFR2 F(ab')₂ and anti-VEGFR2 F(ab')₂-SS31 treated DN mice and were similar with those in the sham group (Fig. 5g–i). Progression of diabetic nephropathy is also characterized by loss of podocyte density, which typically parallels albuminuria[26]. This was quantified by NPHS2, the gene encoding the glomerular protein podocin[27], expressed as a ratio to the mean volume of glomerulus. Podocyte number decreased in DN model mice. After being treated with SS31, there was a slight increase in the number of podocytes per glomerulus compared with nontreated DN mice, and the podocyte number increased obviously in DN model mice administrated with anti-VEGFR2 F(ab')₂ and anti-VEGFR2 F(ab')₂-SS31 (Fig. 5j, k).

To evaluate the anti-oxidative stress effect of anti-VEGFR2 F(ab')₂-SS31 in the kidney of DN, we performed immunohistochemical staining of nitrotyrosine. Nitrotyrosine is formed on protein tyrosine residues by peroxynitrite-induced nitration and is considered a sensitive marker for oxidative stress[28] (Fig. 6a, b). Nitrotyrosine staining was negative in sham mice without DN. The number of nitrotyrosine-positive glomeruli increased significantly in DN mice. SS31, anti-VEGFR2 F(ab')₂, and anti-VEGFR2 F(ab')₂-SS31 treatment decreased the number of nitrotyrosine-positive glomeruli in DN mice. However, anti-VEGFR2 F(ab')₂-SS31 revealed a more obvious reduction of nitrotyrosine caused by DN. The levels of superoxide dismutase (SOD) and

malondialdehyde (MDA), indicators of the extent of injury caused by ROS[21], in kidney tissue were also investigated (Fig. 6c, d). There was a significant depletion of SOD detected in DN kidney tissue. SS31 and anti-VEGFR2 F(ab')₂-SS31 treatment alleviated the depletion of SOD, however, higher SOD level was detected in the anti-VEGFR2 F(ab')₂-SS31 group. Increased MDA levels were found in DN mice. Treatment with SS31 and anti-VEGFR2 F(ab')₂-SS31 decreased the MDA levels in DN-modeled mice.

The accumulation of macrophages is a feature of the development of diabetic nephropathy. Macrophages are one of the central mediators of renal vascular inflammation in diabetes mellitus, and promote diabetic nephropathy through a variety of mechanisms, including production of reactive oxygen species, cytokines, and proteases[25,29]. There was an obvious increase of macrophages in the kidney of the DN group. The treatment of SS31, anti-VEGFR2 F(ab')₂, and anti-VEGFR2 F(ab')₂-SS31 reduced the DN-induced macrophages infiltration, and the number of macrophages in anti-VEGFR2 F(ab')₂-SS31-treated group was similar to sham group without DN (Fig. 6e, f). Apart from the changes in macrophages number, we also tested the alternative in M1/M2 macrophage ratio for assessing DN progress and a precancerous state (Supplementary Fig. 5). It has been reported that M1 macrophage promotes inflammation, and is associated with tissue damage and tumor cell killing[30], whereas M2 macrophage suppresses immune reaction, participates in tissue repair and remodeling, and contributes to proliferation and survival of cancer cells[30,31]. The ratio of M1/M2 macrophages is elevated in the early stage of DN, and decreases as DN progression[32]. The decreased ratio also creates a tumor-promoting microenvironment[33]. In this study, we observed weak signals of M1 and M2 macrophages and a relatively high M1/M2 ratio of ~5.7 in normal mice. In DN mice, the signals of M1 and M2 macrophages increased, and the M1/M2 macrophage ratio decreased to 2.5, indicating DN progression and a more tumor-promoting state. Mice treated with anti-VEGFR2 F(ab')₂ and anti-VEGFR2 F(ab')₂-SS31 had lower macrophage fluorescence and a higher ratio of M1/M2 macrophage (4.4 and 4.4 respectively) than DN mice that were not treated or were administered SS31 (~3.0), indicating that anti-VEGFR2 F(ab')₂ and anti-VEGFR2 F(ab')₂-SS31 effectively inhibited the DN progression and precancerous changing. We further measured the levels of pro-inflammatory cytokines in the kidney tissues, including interleukin-6 (IL-6), tumor necrosis factor-α (TNF-α), interleukin-1β (IL-1β) and interleukin-8 (IL-8)[25,34]. The levels of IL-6, TNF-α, IL-1β, and IL-8 were significantly elevated in kidneys of DN mice. Anti-VEGFR2 F(ab')₂ and anti-VEGFR2 F(ab')₂-SS31 treated mice had lower expression levels of IL-6, TNF-α, IL-1β, and IL-8 than nontreated or SS31 administrated DN mice, and with no difference compared with sham mice, suggesting a strong anti-inflammatory capability (Fig. 6g, h and Supplementary Fig. 6).

Renal fibrosis is a crucial metabolic change in DN, associated with the progression toward end-stage renal disease[35]. We performed immunohistochemical staining of fibrosis factors, including α-smooth muscle actin (α-SMA) and collagen I[36] (Fig. 6i, j and Supplementary Fig. 7a). The expression of fibrosis factors, α-SMA and collagen I, increased in DN kidney and decreased after treatments with anti-VEGFR2 F(ab')₂ and anti-VEGFR2 F(ab')₂-SS31 We further tested the leading cytokines involved in fibrosis, transforming growth factor β1 (TGF-β1)[36] and connective tissue growth factor (CTGF)[37] by ELISA (Supplementary Fig. 7b, c). The levels of TGF-β1 and CTGF obviously increased in DN kidney, and decreased significantly after anti-VEGFR2 F(ab')₂ and anti-VEGFR2 F(ab')₂-SS31 treatments, which was coincident with the results mentioned above, indicating the effective anti-fibrosis ability of anti-VEGFR2 F(ab')₂ and anti-VEGFR2 F(ab')₂-SS31.

## Discussion

Monoclonal antibodies are well-studied in cancer therapy attributed to their targeting effects and high therapeutic efficiency[1], while

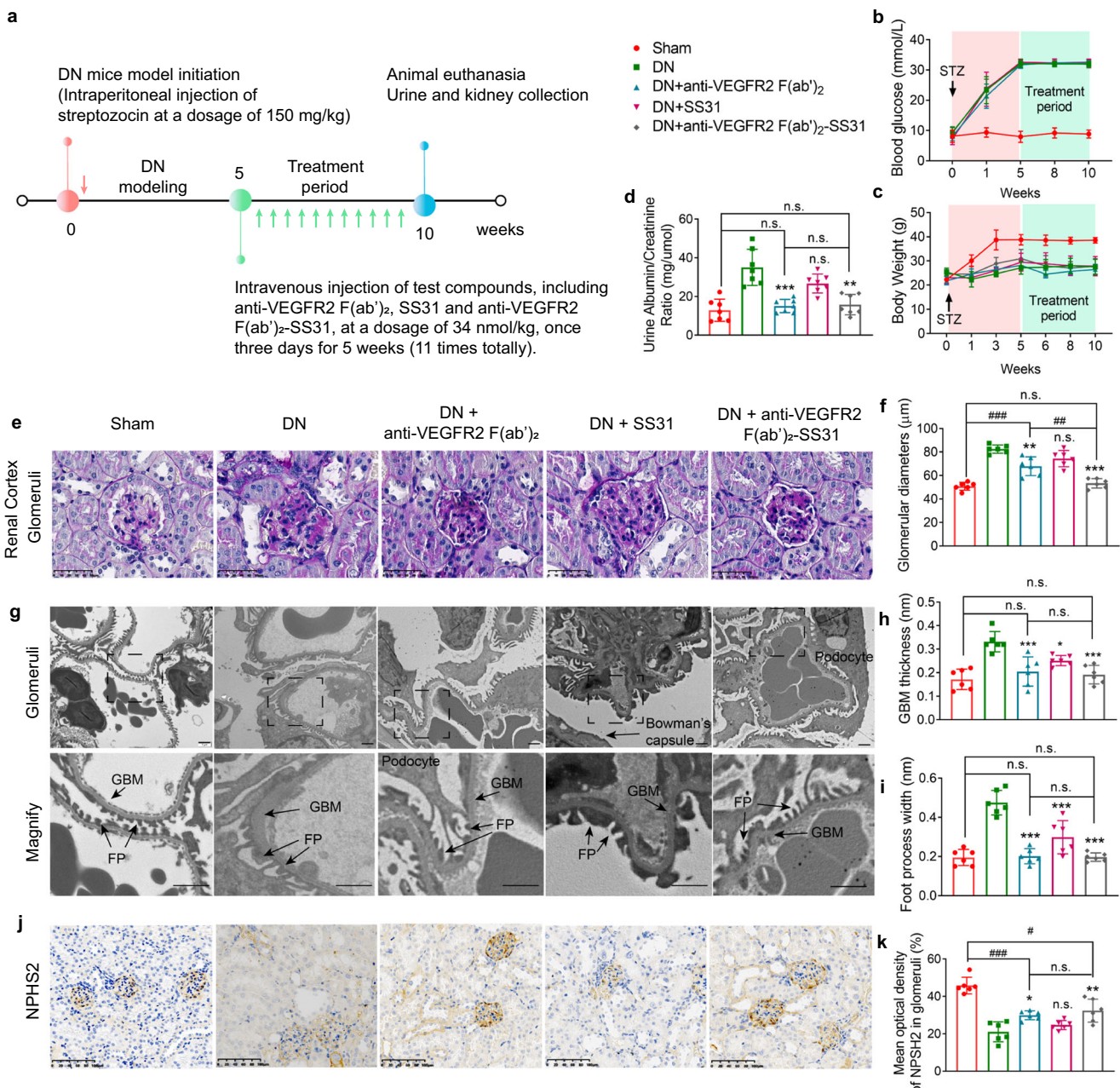

**Fig. 5 | Anti-VEGFR2 F(ab')₂-SS31 treatment protected glomeruli against albuminuria in DN. a** After initiation of streptozotocin-induced DN, different agents (SS31, anti-VEGFR2 F(ab')₂, and anti-VEGFR2 F(ab')₂-SS31) were administered intravenously at a dosage of 33 nmol/kg, once every 3 days for 5 weeks. After administration, urine and kidneys were collected for evaluation. STZ treatment-induced diabetes mice with similar severity, as indicated by (**b**) blood glucose and (**c**) body weight. **d** Urinary albumin/creatinine ratios assessing albuminuria in different groups after different treatments. **e** Representative images of glomeruli (PAS staining of paraffin-fixed sections; scale bar, 50 μm) and dot plots summarizing (**f**) the glomerular diameters. **g** Representative images of the glomerular filtration barrier (transmission electron microscopy; scale bar, 1 μm) and dot plots

summarizing (**h**) the width of the GBM, and (**i**) foot process width, reflecting foot process effacement. **j** Representative photomicrographs of NPSH2 immunohisto-chemical staining. NPSH2 encodes podocin, a slit diaphragm protein in podocytes. Scale bar, 100 μm. **k** Quantitative analysis of NPSH2 staining. All data are expressed as the mean ± s.d. Statistical significance was calculated using a one-way ANOVA and post-hoc test. $n = 6$ mice in each group, n.s. no significant difference, *$p < 0.05$, **$p < 0.01$, ***$p < 0.001$ as compared with the DN group; n.s. no significant difference, #$p < 0.05$, ###$p < 0.001$ between groups as indicated. DN diabetic nephro-pathy, VEGFR2 vascular endothelial growth factor receptor 2, GBM glomerular basement membrane, FP foot process. Source data are provided as a Source Data file.

their application in kidney diseases is limited owing to poor renal distribution. In this study, we found that the renal distribution profile of IgG antibodies is highly influenced by molecular weight. The reduction of molecular weight of anti-VEGFR2 by removing the Fc fragment can effectively increase its renal accumulation and further assist it to block the overactivation of VEGFR2 in DN kidney and inhibit the progress of DN. Fragmentation may be a potential

strategy for the highly effective application of antibody-based drugs in renal diseases.

ADCs have emerged as a potent strategy in cancer therapy, uti-lizing a target of interest as a delivery vehicle for toxic molecules[38]. In this study, our aim was to validate the effectiveness of antibody frag-ments in the development of ADCs for DN therapy. Specifically, we present the development of a new anti-VEGFR2 ADC, called anti-

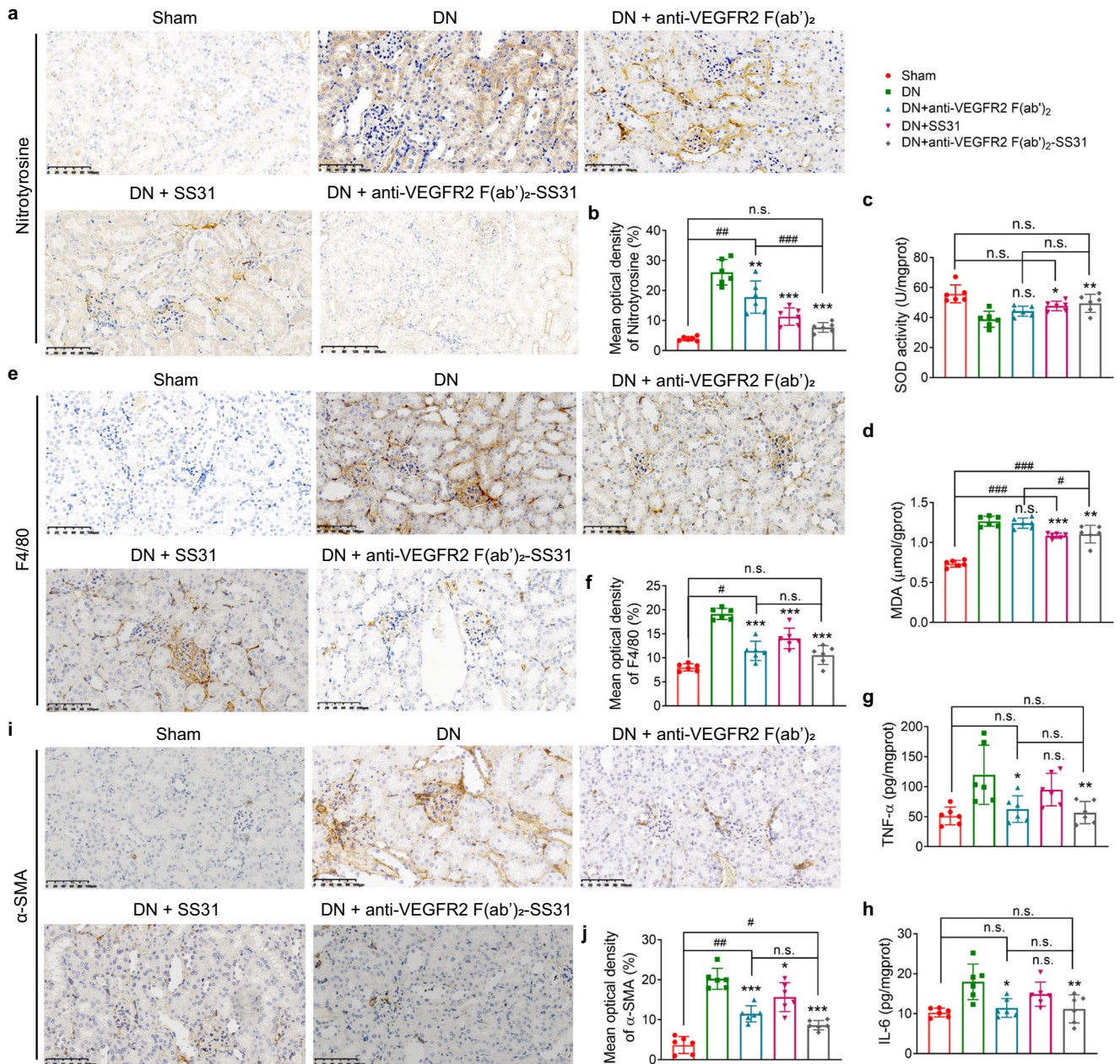

**Fig. 6 | Anti-VEGFR2 F(ab')₂-SS31 reduced oxidative stress, inflammation, and fibrosis in DN mice. a** Immunohistochemical staining (brown) of nitrotyrosine, a marker of peroxidation. Scale bar, 100 µm. **b** Quantitative analysis of nitrotyrosine staining in **a. c, d** SOD and MDA level changes in DN mice after different treatments. **e** Representative photomicrographs of F4/80 staining, a marker of macrophage. Scale bar, 50 µm. **f** Quantitative analysis of F4/80 staining. **g, h** Renal cytokine TNF-α and IL-6 alternation in DN mice after different treatments. **i** Representative photomicrographs of α-SMA, a marker of fibroblast. Scale bar, 50 µm. **j** Quantitative analysis of α-SMA expression. All data are expressed as the mean ± s.d. Statistical significance was calculated using a one-way ANOVA and post-hoc test. $n = 6$ mice in each group, n.s. no significant difference, $*p < 0.05$, $**p < 0.01$, $***p < 0.001$ as compared with DN group; n.s. no significant difference, $\#p < 0.05$, $\#\#p < 0.01$, $\#\#\#p < 0.001$ between groups as indicated. DN diabetic nephropathy, VEGFR2 vascular endothelial growth factor receptor 2, SOD superoxide dismutase, MDA malondialdehyde, TNF-α tumor necrosis factor-α, IL-6 interleukin-6, α-SMA α-smooth muscle actin. Source data are provided as a Source Data file.

VEGFR2 F(ab')₂-SS31, which incorporated the antioxidant peptide SS31 as a payload. This choice was based on the recognition that oxidative stress plays a crucial role in the progression of DN[11,20]. We demonstrated that fragmentation and SS31 conjugation did not weaken the affinity of anti-VEGFR2 to VEGFR2. Moreover, both anti-VEGFR2 F(ab')₂ and anti-VEGFR2 F(ab')₂-SS31 could efficiently distribute within the cytoplasm of MRGEC and MPC5 cultured in a high glucose medium, mimicking the conditions of DN. Our investigation indicated that anti-VEGFR2 F(ab')₂ exerted inhibitory effects on the permeability and migration of VEGF-induced glomerular endothelial cells by suppressing the VEGF/VEGFR2 pathway. Moreover, anti-VEGFR2 F(ab')₂

exhibited potent therapeutic effects in DN mouse models, including protection of glomerular structure, and reduction of pro-inflammatory factors and fibrosis indices. Notably, anti-VEGFR2 F(ab')₂-SS31, in addition to the outstanding performance in the above aspects, confers the advantage of attenuating oxidative stress due to its SS31 conjugation. (Fig. 7). In conclusion, the renal accumulation and excellent therapeutic response in DN mice suggests that anti-VEGFR2 F(ab')₂-SS31 is a promising candidate for the treatment of clinical DN. The application of antibody-based drugs and ADCs in the therapy of renal diseases remains a field with great potential for exploration in the future.

**a** Renal accumulation of anti-VEGFR2 F(ab')₂-SS31
after intravenous administration to DN mouse

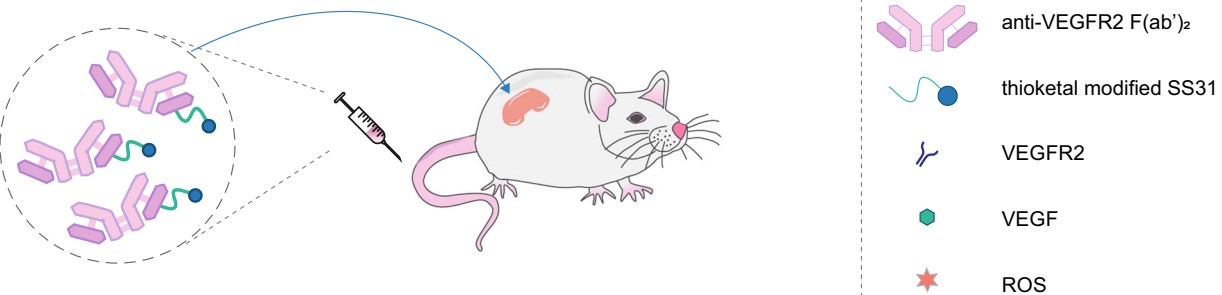

**b** Anti-VEGFR2 F(ab')₂-SS31 promotes glomerular repair

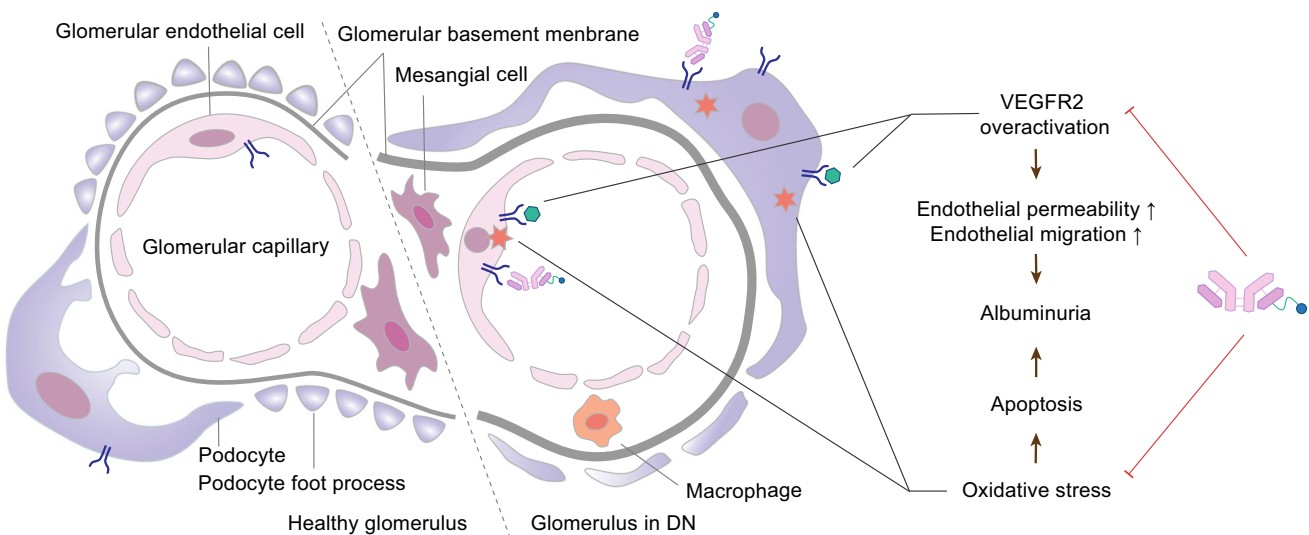

**Fig. 7 | Schematic illustration of relieving DN by anti-VEGFR2 F(ab')₂-SS31.** After intravenous administration, anti-VEGFR2 F(ab')₂-SS31 accumulates in kidneys subject to DN. The overexpressed VEGFR2 in glomerular cells (including glomerular endothelial cells and podocytes) in DN is associated with the internalization of anti-VEGFR2 F(ab')₂-SS31. Anti-VEGFR2 F(ab')₂-SS31 further repairs glomeruli by inhibiting VEGFR2 overactivation and decreasing oxidative stress, ultimately reducing albuminuria and alleviating DN. VEGFR2 vascular endothelial growth factor receptor 2, DN diabetic nephropathy, VEGF vascular endothelial growth factor, ROS reactive oxygen species. Reproduced with permission from ref. 8. Copyright 2021, American Chemical Society.

## Methods

### Generation of anti-VEGFR2 F(ab')₂

Anti-VEGFR2 F(ab')₂ fragments were generated by pepsin degradation. Briefly, anti-VEGFR2 monoclonal antibodies (anti-VEGFR2) (7 mg/mL, 1 mL, DC101, #BP0060, Bio X Cell) in acetic acid-sodium acetate buffer (pH 3.4) were incubated with pepsin (Sigma) with the mass ratio of 20:1 for 6 h at 37 °C, followed by purification on ultrafiltration centrifuge tube (MWCO 30 kD, 0.5 mL, Millipore), centrifuging at 14000 × g for 15 min, and washing with PBS (pH 7.4) three times to remove small peptides of Fc fragments. Collect anti-VEGFR2 F(ab')₂, adjust the volume, and measure the concentration by Nanodrop One (Thermo Scientific). Rat immunoglobulin G (IgG, Shanghai yuanye) was operated in the same way, leading to IgG F(ab')₂ as an isotype control antibody. Measure the mass of TCEP (Aladdin) reduced anti-VEGFR2 F(ab')₂ by ultraflex TOF/TOF mass spectrometry (Bruker) from 10 to 60 kDa.

### Synthesis of NHS ester-modified thioketal linker (NHS-TK-NHS)

First, synthesize the ROS-cleavable thioketal linker (TK) according to previous reports[16,39]. Then, synthesize NHS ester-modified TK (NHS-TK-NHS)[40]. Briefly, TK (112 mg, 448 µmol), 1-Ethyl-3-(3-dimethylaminopropyl)carbodiimide hydrochloride (EDCI) (205 mg, 1.07 mmol, Aladdin), and N-Hydroxy succinimide (NHS) (123 mg, 1.07 mmol,

Aladdin) were successively added into DCM (50 mL). After stirring overnight at room temperature, the reaction mixture was washed twice successively with 1.0 M HCl, saturated NaHCO₃, and saturated NaCl. The organic phase was then collected and dried over anhydrous MgSO₄, followed by filtering. Finally, the solvent was removed using a rotary evaporator and NHS-TK-NHS was obtained. The structure of NHS-TK-NHS was identified using ¹H-NMR spectroscopy (AC-80, Bruker) in d₆-DMSO (Supplementary Fig. 8).

### Synthesis of anti-VEGFR2 F(ab')₂-SS31

Aliquot SS31 (10 mg/mL, 18 µL, 290 nmol, Popchem) in PBS (0.01 M, pH 7.4) into a 0.5 mL Eppendorf tube, and add NaHCO₃ (1 M, 1.8 µL) to adjust the solution alkaline. NHS-TK-NHS was added as a DMSO solution (33 mg/mL, 3 µL, 220 nmol). The reaction was incubated in a shaker for 15 min at room temperature. Then, anti-VEGFR2 F(ab')₂ (7.3 mg/mL, 0.1 mL, 66 nmol) PBS solution and NaHCO₃ (1 M, 1.8 µL) were added. After 30 min of incubation, lysine (10 mg/mL, 6.4 µL, 440 nmol) was added to block the remaining reactive carboxylic groups. The reaction mixture was incubated for another 1 h at room temperature. Unbound SS31, synthetic by-products, and lysine were removed by repetitive centrifugation (14,000×g, 10 min at a time, 3 times) with an ultrafiltration centrifuge tube (MWCO 30 kD, 0.5 mL, Millipore). Then SS31-conjugated anti-VEGFR2 F(ab')₂ (anti-VEGFR2 F(ab')₂ -SS31) was

synthesized. Collect anti-VEGFR2 F(ab')$_2$-SS31, adjust the volume with PBS, and measure the concentration by Nanodrop One. Measure the mass of TCEP reduced anti-VEGFR2 F(ab')$_2$-SS31 by mass spectrometry from an 18-30 kDa, and calculate DAR using Eq. (1).

$$DAR = 4 \times \frac{\text{Peak Area of Double Drug Conjugated LC} \times 2 + \text{Peak Area of Triple Drug Conjugated LC (3)} \times 3}{\text{Peak Area of Unconjugated LC} + \text{Peak Areas of Conjugated LC}} \qquad (1)$$

### Preparation of fluorescence-labeled anti-VEGFR2 F(ab')$_2$

Anti-VEGFR2 F(ab')$_2$ (47 μM, 250 μL) in PBS was mixed with Cy5-NHS (13 mM, 1 mL, Dalian Meilun) and NaHCO$_3$ (1 M, 25 μL), and incubated in a shaker at room temperature for 2 h. Unbound dye molecules were removed by repetitive centrifugation (14,000 × g, 10 min at a time, 3 times) with an ultrafiltration centrifuge tube (MWCO 30 kD, 0.5 mL, Millipore) until no fluorescence was detected in the supernatant. Anti-VEGFR2, IgG, IgG-F(ab')$_2$, and anti-VEGFR2 F(ab')$_2$-SS31 were operated in the same way as the control groups.

### DN mouse model and biodistribution of anti-VEGFR2 F(ab')$_2$

ICR mice (male, 6–8 weeks old, 20–25 g) were purchased from Zhejiang Academy of Medical Sciences (Hangzhou, China). The mice were housed in conventional conditions with standard food and water. The mice were housed at approximately 22 ± 2 degrees centigrade, humidity 50 ± 10% on a 12 h light/12 h dark cycle. The chow was purchased from Puluteng Feed (Slacom P1101F, China), consisted of fish meal, wheat, and corn. The purified drinking water was purchased from Wahaha (596 purified drinking water, China). All animal experiments were performed following the National Institutes of Health Guide for the Care and Use of Laboratory Animals with the approval of the Scientific Investigation Board of Zhejiang University, Hangzhou, China, under production license number ZJU20230213.

The experimental protocol of DN model mice was described in previous reports[41]. Mice were fasted overnight and intraperitoneally injected with STZ (150 mg/kg, dissolved in 50 mM pH 4.5 citrate buffer solution); the mice in the control group were injected intraperitoneally with citrate buffer. A week later, mice were fasted overnight and the fasting blood glucose (FBG) was tested by using a One Touch Basic Blood Glucose Monitoring Apparatus. Diabetes was defined as FBG ≥ 15 mmol/L. The body weight and FBG of the mice were monitored. The worsening of diabetes symptoms or the occurrence of serious complications was a sign of diabetes developing into DN at 5 weeks after STZ treatment.

To determine the biodistribution behavior of anti-VEGFR2 F(ab')$_2$ in DN kidney, mice at 5 weeks after STZ treatment were randomly divided into (1) the IgG-Cy5 group, in which the DN mice were given IgG-Cy5 (0.2 mL, at a dose of 5 mg/kg) intravenously; (2) the IgG F(ab')$_2$-Cy5 group, in which the DN mice were given IgG-F(ab')$_2$-Cy5 (0.2 mL, at a dose of 3.7 mg/kg) intravenously; (3) the anti-VEGFR2-Cy5 group, in which the mice were subjected to DN and then given anti-VEGFR2-Cy5 (0.2 mL, at a dose of 5 mg/kg) intravenously; and (4) the anti-VEGFR2 F(ab')$_2$-Cy5 group, in which the mice were subjected to DN and then given anti-VEGFR2 F(ab')$_2$-Cy5 (0.2 mL, at a dose of 3.7 mg/kg) intravenously. At 4 h, 24 h, or 48 h after administration, euthanasia was administered to the mice by overdose injection of pentobarbital sodium, and main organs including heart, lung, liver, spleen, and kidney were harvested for fluorescence visualization. Healthy mice were administrated and processed as described above. The organs were visualized by Maestro in vivo imaging system (IVIS Spectrum, PerkinElmer), and the relative fluorescent intensity was calculated in the same system.

### SDS-PAGE gels

Non-reducing glycine-SDS-PAGE at 6%, 8%, or 12% acrylamide gels was performed following standard laboratory procedures. A 6%, 8%, or 12% separation gel was used and a broad-range molecular weight marker (10-170 kDa, Beyotime) was co-run to estimate protein weights. Intact and fragmented antibodies (anti-VEGFR2, anti-VEGFR2 F(ab')$_2$, IgG, anti-VEGFR2 F(ab')$_2$-SS31) were collected and analyzed by SDS-PAGE. Samples (100 μL at ~10 mM in total antibody) were mixed with loading buffer (5×, Beyotime) and heated at 100 °C for 5 min, then loaded on the gels. After electrophoresis, all gels were stained with Feto SDS-PAGE staining buffer (Absci).

### The binding ability of mouse VEGFR2

Mouse VEGFR2 (0.25 mg/mL, 50 μL, Sino Biological) was mixed with loading buffer (5×, Beyotime) and heated at 100 °C for 5 min. Western blotting was performed with standard techniques using 8% acrylamide gels and polyvinylidene fluoride (PVDF) membrane (Merck Millipore). Blots were blocked in 4% milk in TPBS and incubated with primary antibodies: 1) anti-VEGFR2 (6 μg/mL, BioCell InvivoPlus), 2) anti-VEGFR2 F(ab')$_2$ (4.4 μg/mL), 3) anti-VEGFR2 F(ab')$_2$-SS31 (4.4 μg/mL), and rat IgG (6 μg/mL, Nanjing Feiyu) as a negative control. The following secondary antibodies were used: Goat Anti-Rat IgG (F(ab')$_2$ antibody peroxidase conjugate (1:2000, GeneTex, GTX26517). The secondary antibody was imaged using the Chemical illuminant (BeyoECL Plus) and quantified in the Bio-Rad system (ChemiDoc Touch Imaging System). The experiments were repeated thrice.

### Cell lines

Mouse renal glomerular endothelial cells (MRGECs) and mouse podocyte clone5 (MPC5) cells were purchased from Jennio Biotech (GuangZhou, China) with catalog numbers of JNO-M0021 and JNO-M0229, respectively. Both cells were maintained at 37 °C, 5% CO$_2$ in low glucose medium (Dulbecco's modified Eagle's medium (DMEM) with 5.6 mM glucose, Solarbio, China) complemented with 10% fetal serum (FBS) (Procell, China) and 100 U/mL penicillin/streptomycin (Solarbio, China). High glucose-stimulated model cells were incubated for 24 h in high glucose medium (DMEM, 10% FBS, mediated glucose concentration to 30 mM using D-glucose).

### VEGFR2 expression in cells or tissues

MRGECs or MPC5 cells were lysed according to the lysis protocol in ice-cold RIPA. The kidney tissues were homogenized in ice-cold phosphate buffer (10 mM; pH 7.4). Collect supernatant after centrifugation (13,000 ×g for 5 min). The concentration of protein in the supernatant was tested using a BCA-protein assay kit (Beyotime). Supernatant samples were mixed with loading buffer (5×, Beyotime) and heated at 100 °C for 5 min. Western blotting was performed with standard techniques using 8% acrylamide gels and PVDF membranes. Blots were blocked in 4% milk in TPBS and incubated with primary antibodies: VEGF Receptor 2 (D5B1) Rabbit mAb (1:1000, Cell signaling); β-actin (1:2500, Proteintech). The following secondary antibody were used: goat anti-rabbit immunoglobulin G (IgG) horseradish peroxidase (1:1000, Beyotime). The secondary antibody was imaged using the Chemical illuminant (BeyoECL Plus) and quantified in the Bio-Rad system (ChemiDoc Touch Imaging System).

### Internalization analysis by confocal microscopy

MRGECs and MPC5 cells on coverslips were stimulated with high glucose (30 mM) for 24 h, and incubated with Cy5-labeled constructs,

including anti-VEGFR2-Cy5, anti-VEGFR2 F(ab')$_2$-Cy5, and anti-VEGFR2 F(ab')$_2$-SS31-Cy5 at 10 mg/mL for 1 h at 4 °C or 37 °C. MRGECs and MPC5 cells incubated in low glucose mediums were taken as controls and operated in the same way. Then the cells were washed extensively and fixed with 4% formaldehyde for 10 min. Coverslips were then blocked with 5% goat serum. Hoechst trihydrochloride (Beyotime) was used to stain cell nuclei. Coverslips were mounted on slides and examined using CLSM (Leica TCS SP8). The experiments were repeated thrice.

### Transwell migration assay
The motility of MRGECs was performed in 24-well transwell plates using 8 μm polycarbonate filters. In all, $1 \times 10^5$ cells suspended in 200 μL 10% FBS high glucose DMEM were seeded into the upper chamber, and the bottom chamber was filled with 600 μL 20% FBS high glucose DMEM medium. VEGF (10 ng/mL) with and without different samples (anti-VEGFR2 F(ab')$_2$, SS31, or anti-VEGFR2 F(ab')$_2$-SS31 at a concentration of 0.4 μM) were added to the upper chambers at the same time and incubated for 24 h. After that, the cells were fixed with 4% paraformaldehyde for 10 min and stained with 0.1% crystal violet. The migrated cells on the lower surface were photographed with amicroscope (DMIL, Leica) in five randomly selected visual fields and the migrated cells were quantified. The experiments were repeated thrice.

### Scratch assay
MRGECs were placed into 12-well plates at a density of $2 \times 10^4$ cells/well for overnight incubation. Then, uniform scratches were made in the cells. VEGF (10 ng/mL) with and without different samples (anti-VEGFR2 F(ab')$_2$, SS31, or anti-VEGFR2 F(ab')$_2$-SS31 at a concentration of 0.4 μM) were added simultaneously. At 0 and 12 h, the migration distance was imaged by a light microscope and quantified by ImageJ software. The experiments were repeated thrice.

### Mitochondrial ROS detection
MRGECs and MPC5 cells were seeded into 12-well plates and incubated with high glucose medium for 24 h. Different samples (anti-VEGFR2 F(ab')$_2$, SS31, or anti-VEGFR2 F(ab')$_2$-SS31 at a concentration of 0.4 μM) were added and incubated for another 24 h. After washing with PBS three times, cells were incubated with MitoSOX Red Mitochondrial Superoxide Indicator (5 μM) for 10 min. Thereafter the mitochondrial ROS level was determined by CLSM. Cells incubated with low glucose or high glucose medium without drug treatment were used as the control groups. The experiments were repeated thrice.

### Apoptosis detection in vitro
MRGECs and MPC5 cells were seeded into 12-well plates and incubated with high glucose medium for 24 h. Different samples (anti-VEGFR2 F(ab')$_2$, SS31, or anti-VEGFR2 F(ab')$_2$-SS31 at a concentration of 0.4 μM) were added and incubated for another 24 h. Then, cells were harvested, resuspended in the buffer, incubated with annexin V-fluorescein isothiocyanate, stained with propidium iodide (Beyotime), and examined using a flow cytometer (ACEA NovoCyteTM; ACEA Biosciences). Flow cytometry was repeated in three independent experiments.

### Administration protocols for in vivo therapeutic evaluation
The DN mice were intravenously injected with anti-VEGFR2 F(ab')$_2$ (3.7 mg/kg), SS31 (22 μg/kg), or anti-VEGFR2 F(ab')$_2$-SS31 (3.7 mg/kg) at a dosage of 34 nmol/kg (6 mice in each group). The sham group and DN group were treated intravenously with saline (6 mice in each group). The administration was performed every three days for 5 weeks. The fasting blood glucose (FBG) and body weight of mice were measured and recorded. At the end of the treatment, urine was collected for detection of albuminuria. Mice were sacrificed and kidneys were harvested for histologic analysis.

A one-time high-dose intraperitoneal injection of STZ (150 mg/kg) rapidly damaged pancreatic islet cells, leading to hyperglycemia. After administration of STZ for 5 weeks, the anti-VEGFR2 F(ab')$_2$, SS31, and anti-VEGFR2 F(ab')$_2$-SS31 were administrated by intravenous injection once every 3 days for 5 weeks at a single dosage of 34 nmol/kg, followed by collecting the urine and kidneys 48 h later for further evaluation

### Renal function and histology
Urine was tested by urine creatinine and urine microalbumin kits (Shanghai mlbio). Kidney histology was examined in formalin-fixed, periodic acid-Schiff stain (PAS)-stained sections. Observe the renal lesions under an optical microscope and measure glomerular diameters.

### Oxidative stress and pro-inflammatory cytokines assay
The levels of SOD and MDA were examined using commercial kits (SOD: Beyotime, MDA: Nanjing Jiancheng) according to the manufacturer's instructions. The levels of TNF-α, IL-6, IL-1β, IL-8, TGF-β1, and CTGF were quantified using ELISA kits (Jiangsu Meimian industrial Co., Ltd) according to the manufacturer's procedures.

### Electron microscopy
Renal tissues were fixed in 2.5% glutaraldehyde, fixed in 1% osmium tetroxide, dehydrated in graded alcohols, and embedded in epoxy resin. Ultrathin sections were cut and stained with uranyl acetate and lead citrate, and examined by transmission electron microscope (TEM, JEOL JEM-1230).

### Immunohistochemical staining
Deparaffinized and rehydrated renal tissue sections were blocked in blocking buffer for 30 min at room temperature, incubated with anti-NPHS2 (1:400, Abcam, ab181143), anti-nitrotyrosine (3 μg/mL. Gene-Tex, GTX30730), or anti-F4/80 ((1:400, Cell Signaling, 70076s), or anti-α-SMA (1:500, Proteintech, 14395-1-AP), or anti-collagen I (1:1000, Proteintech, 14695-1-AP), and incubated with HRP-conjugated rabbit anti-mouse IgG secondary antibody. The mean optical density of NPHS2, nitrotyrosine, F4/80, or α-SMA was counted from five different fields of each sample.

### Immunofluorescence staining
The kidneys of mice were acquired 4 h or 24 h after intravenous injection of anti-VEGFR2-Cy5 or anti-VEGFR2 F(ab')$_2$-Cy5. Kidney sections were exposed to the primary antibody (1:1000, VEGF Receptor 2 (D5B1) Rabbit mAb, Cell Signaling), then to FITC-labeled secondary antibody (1:1000, Beyotime). Renal sections were also treated with 2-(4-amidinophenyl)-1H-indole-6-carboxamidine (DAPI, Beyotime). The staining was visualized using fluorescence microscopes.

Immunofluorescence staining of renal tissues was performed for labeling M1/M2 macrophage. Deparaffinized and rehydrated renal tissue sections were blocked in blocking buffer for 30 min at room temperature, sequentially incubated with anti-CD86 (1:400, Cell Signaling, 19589 s) and anti-CD206 (1:1000, Abcam, ab300621) along with fluorescently-labeled secondary antibodies. Renal sections were also treated with DAPI. The staining was examined using fluorescence microscopes.

### Statistical data analysis
All data were analyzed using GraphPad Prism 8.0.2. The results are expressed as the mean ± s.d. To compare the significant differences, a one-way ANOVA or two-tailed unpaired $t$ test was used, considering the $p$-value to be 0.05 or less.

**Reporting summary**

Further information on research design is available in the Nature Portfolio Reporting Summary linked to this article.

## Data availability

All data associated with this study are available within the Article, Supplementary Information, or Source data file. Source data are provided with this paper.

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

## Acknowledgements

This work was supported by the Zhejiang Provincial Natural Science Foundation (No. LD21H300002. Y.Z.D.) and National Natural Science Foundations of China (82170725. S.P.J.).

## Author contributions

Y.Z.D., S.P.J., and X.L.X. designed and guided the overall research project. D.L. designed the experiments and wrote the manuscript. Y.L.S. performed the actuation experiments. H.C. and Y.C.Y. were involved in the synthesis of ADCs and cellular experiments. L.W.Z. and J.C.Z. assisted with animal maintenance. X.Y.X. and J.H.H. were involved in the data analysis and other property characterizations. X.J.H. and X.C.W. provided intellectual input and helped interpret the results.

## Competing interests

The authors declare no competing interests.
