## [Peer Review File · Nature Communications]

Anti-VEGFR2 F(ab')₂ drug conjugate promotes renal accumulation and glomerular repair in diabetic nephropathyREVIEWER COMMENTS

Reviewer #1 (Remarks to the Author):

The work of Liu D et al is dedicated to the treatment of diabetic nephropathy (DN). This pathology is often at an impasse and requires a better treatment option.

DN is characterised by abnormal angiogenesis and increased VEGF/VEGFR2 signalling.

Therefore, Liu and colleagues investigated the importance of anti-VEGFR2 (F(ab')₂) antibody for the treatment of DN. In addition, and this is a clever method, they used ADC antibody with the antioxidant peptide SS31 as a P-load.

They first demonstrated that anti-VEGFR2 (F(ab')₂) antibodies were distributed more efficiently in the kidney of DN mice compared to healthy mice, but more importantly as full-length anti-VEGFR2 antibodies, which confirmed the proof of concept of using smaller antibodies.

Liu and al. then compared the in vitro effects of the anti-VEGFR2 antibody, the anti-VEGFR2 (F(ab')₂) antibody and the anti-VEGFR2 (F(ab')₂) antibody coupled to the SS31 peptide. The three antibodies recognise VEGFR2 with the same affinity.

They showed that the anti-VEGFR2 (F(ab')₂) antibodies coupled to the SS31 peptide were more efficiently distributed in the cytoplasm of mouse glomerular kidney endothelial cells (MRGEC) and mouse podocytes (MPC5) cultured in a high glucose medium mimicking DN.

Under high glucose conditions, anti-VEGFR2 (F(ab')₂) and anti-VEGFR2 (F(ab')₂)-SS31 inhibited VEGF-dependent migration, oxidative stress and apoptosis of MRGECs.

Moreover, anti-VEGFR2 (F(ab')₂)-SS31 protected glomeruli from albuminuria in STZ mice more effectively than anti-VEGFR2 (F(ab')₂).

A more detailed analysis showed that anti-VEGFR2 (F(ab')₂)-SS31 reduced nitrotyrosine levels SOD and infiltration of macrophages and reduced levels of the inflammatory cytokines IL6 and TNF.

The results of Liu et al are of primary quality and of first importance for the treatment of DN and deserve to be published in Nat Comm if they can provide the additional results/precision/methodology.

Important points:

1-The origin of the anti-VEGFR2 antibody used in this study is not clear. The author needs to

describe this in detail. It is of course not clear if these antibodies have already been used in the clinic (associated patent) and for which pathology.

2-The signalling pathways inhibited by the different antibodies should have been tested at least in vitro and if possible in the kidney of the treated mice (the latter is not mandatory).

3-In most in vivo experiments, the anti-VEGFR2 (F(ab')₂) and the anti-VEGFR2 (F(ab')₂)-SS31 showed equivalent results, although the author argues that the anti-VEGFR2 (F(ab')₂)-SS31 has greater efficacy. The text should be revised with this in mind.

4-In the results shown in Figure 6, the author has evaluated the levels of IL6 and TNF. There are other inflammatory cytokines that should have been tested for a larger panel, including interleukin 8, at least IL1beta.

5-The author describes the presence of macrophages in general. However, did they test the relative M1/M2 macrophage ratio to possibly assess a precancerous state?

6-The authors address fibrosis, which is very important considering that DN can cause dead-end fibrosis. Other fibrosis markers should have been tested at least by qPCR (FAP, collagen, ...).

7-A very important parameter in this context is the fibrosis parameter. Among the cytokines involved in this parameter, TGF beta, CCN2/CTGF are leading cytokines that should be tested.

8-There are several anti-angiogenics that have been approved in the clinic for the treatment of abnormal angiogenesis. How do the anti-VEGFR2 (F(ab')₂)-SS31 antibodies compare to the historical anti-VEGF Avastin/bevacizumab and/or multi-tyrosine kinase inhibitors, at least on cultured cells.

Minor concerns:

1-The following sentence in the abstract is unclear: "We found that the anti-VEGFR2 F(ab')₂ had higher accumulation in DN mouse kidneys and comparable VEGFR2 combination with the intact VEGFR2 antibody."

2-The last sentence of the introduction is too long.

3-The "Conclusion/Discussion" part is very short and should be expanded to put the results in the general context of treating DN.

Reviewer #2 (Remarks to the Author):

Overview and general recommendation:

This study investigates the viability of utilizing monoclonal antibodies for renal therapy. In particular, it offers a new way to treat the disease by making anti-VEGFR2 F(ab')₂-SS31 gather together in the kidney. By removing the Fc fragment and conjugating SS31 to anti-VEGFR2, this is possible. VEGFR2 F(ab')₂-SS31 can exert inhibitory effects on cell migration, oxidative stress, and apoptosis through experimental validation conducted in vitro. Hence, I will now provide a more comprehensive explanation of my suggestions.

1. It is recommended to provide additional details regarding the rationale behind the choice of the mitochondria-targeted antioxidant peptide SS31 in in vitro and in vivo experiments.
2. If possible, it would be good to do more tests to show that anti-VEGFR2 F(ab')₂-SS31 has the potential to reduce proteinuria and blood creatinine levels in diabetic mice and even in humans.
3. There needs to be more talk about the important role of the anti-VEGFR2 F(ab')₂ fragment in renal therapy using SS31 technology in the conclusion section.
4. There exist a few minor errors, such as inconsistencies in the style of references, which necessitate thorough scrutiny.
5. The improvement of English language proficiency is necessary.

Reviewer: 1

The work of Liu D et al is dedicated to the treatment of diabetic nephropathy (DN). This pathology is often at an impasse and requires a better treatment option. DN is characterised by abnormal angiogenesis and increased VEGF/VEGFR2 signalling. Therefore, Liu and colleagues investigated the importance of anti-VEGFR2 (F(ab')₂) antibody for the treatment of DN. In addition, and this is a clever method, they used ADC antibody with the antioxidant peptide SS31 as a P-load. They first demonstrated that anti-VEGFR2 (F(ab')₂) antibodies were distributed more efficiently in the kidney of DN mice compared to healthy mice, but more importantly as full-length anti-VEGFR2 antibodies, which confirmed the proof of concept of using smaller antibodies. Liu and al. then compared the in vitro effects of the anti-VEGFR2 antibody, the anti-VEGFR2 (F(ab')₂) antibody and the anti-VEGFR2 (F(ab')₂) antibody coupled to the SS31 peptide. The three antibodies recognise VEGFR2 with the same affinity. They showed that the anti-VEGFR2 (F(ab')₂) antibodies coupled to the SS31 peptide were more efficiently distributed in the cytoplasm of mouse glomerular kidney endothelial cells (MRGEC) and mouse podocytes (MPC5) cultured in a high glucose medium mimicking DN. Under high glucose conditions, anti-VEGFR2 (F(ab')₂) and anti-VEGFR2 (F(ab')₂)-SS31 inhibited VEGF-dependent migration, oxidative stress and apoptosis of MRGECs. Moreover, anti-VEGFR2 (F(ab')₂)-SS31 protected glomeruli from albuminuria in STZ mice more effectively than anti-VEGFR2 (F(ab')₂). A more detailed analysis showed that anti-VEGFR2 (F(ab')₂)-SS31 reduced nitrotyrosine levels

SOD and infiltration of macrophages and reduced levels of the inflammatory cytokines IL6 and TNF. The results of Liu et al are of primary quality and of first importance for the treatment of DN and deserve to be published in Nat Comm if they can provide the additional results/precision/methodology.

Response: Thank the reviewer for the positive comments and consideration “The results are of primary quality and of first importance for the treatment of DN”. We appreciate receiving your kind and valuable comments. We have studied the comments carefully and have made corrections which we hope to meet with approval.

All the responses here are in blue. All the changes have been highlighted by giving the text a yellow background in the revised manuscript.

Important points:

1. Response to comment: The origin of the anti-VEGFR2 antibody used in this study is not clear. The author needs to describe this in detail. It is of course not clear if these antibodies have already been used in the clinic (associated patent) and for which pathology.

Response: We thank the reviewer for raising the inquiries about the specifics of the antibody we employed in our research. The antibody utilized in our study was purchased in Bio X Cell (Product name: *InVivoPlus* anti-mouse VEGFR-2, Catalog: #BP0060, Clone: DC101, Isotype: Rat IgG1 κ , Immunogen: Mouse VEGFR-2-SEAPs soluble receptor). Detailed information, including the aforementioned product name, catalog, and clone, can be found in Line 342-343 of the revised manuscript.

It is important to note that *InVivoPlus* anti-mouse VEGFR-2 is just for laboratory use and is not intended for any clinical, therapeutic, or diagnostic use in humans or animals. The primary application of this product involves *in vivo* blocking VEGF/VEGFR-2 signaling (in mouse tumor models and diabetic models) to inhibit vascular proliferation and permeability and thereby suppressing cancer and reverse diabetes [Cancer Res. 2015, 75, 330-343; Cancer cell 2015, 28, 210-224; Diabetes 2013, 62, 2870-2878]. In our study, we capitalized on its capacity to block VEGFR-2 overexpressed in the kidneys of mice suffering from diabetic nephropathy [Kidney Int. 2010, 77, 989-999; J. Pathol. 2010, 226, 562-574]. This was also the solid background that we could further explore VEGFR-2 blocking ability and therapeutic effects of the fragmented antibody in the context of diabetic nephropathy. The revised manuscript provided the information regarding origin and therapeutic the mechanism of anti-VEGFR2 in Line 75-76.

2. Response to comment: The signalling pathways inhibited by the different antibodies should have been tested at least *in vitro* and if possible in the kidney of the treated mice (the latter is not mandatory).

Response: Thank you for your kind and valuable advice. It has helped us enhance the clarity of the signaling pathways inhibited by the antibodies we employed in the study. All the antibody products utilized in the research, namely anti-VEGFR2, anti-VEGFR2 F(ab')₂, and anti-VEGFR2 F(ab')₂-SS31, are derived from anti-VEGFR2 antibody procured from Bio X Cell, specifically the clone number DC101. The DC101

monoclonal antibody reacts with mouse VEGFR2, also known as CD309, KDR, and Flk-1. VEGFR2 is a member of the tyrosine protein kinase family and plays key roles in vascular development and permeability upon binding to its ligand VEGF [Nat. Med. 2003, 9, 669–676; Nat. Rev. Mol. Cell. Biol. 2006, 7, 359–371]. *In vivo* studies have demonstrated the inhibitory effects of the DC101 antibody on VEGF/VEGFR2 signaling [Cancer Res. 2015, 75, 330-343; Cancer cell. 2015, 28, 210-224; Diabetes. 2013, 62, 2870-2878]. VEGF induces phosphorylation of VEGFR-2 and the activity of VEGFR-2 mediated signaling cascades [Eur. J. Pharmacol. 2011, 667, 153–159]. To evaluate the blocking effects of the antibodies employed in the study on the VEGF/VEGFR2 pathway, we examined the expression of phosphorylated VEGFR2 in the kidney of mice with DN following different treatments (**Fig. R1**). The results demonstrated a decrease in the levels of phosphorylated VEGFR-2 (p-VEGFR2) in DN mice after administration of the antibody products, including anti-VEGFR2, anti-VEGFR2 F(ab')₂, and anti-VEGFR2

Fig. R1. Representative photomicrographs of p-VEGFR2 staining in DN kidneys after administration of anti-VEGFR2, anti-VEGFR2 F(ab')₂, and anti-VEGFR2 F(ab')₂-SS31. Scale bar: 100 μ m.

F(ab')₂-SS31. This observation indicated the ability of these antibodies to block the

VEGF/VEGFR2 pathway.

Furthermore, we investigated the inhibitory capability of anti-VEGFR2 F(ab')₂ and anti-VEGFR2 F(ab')₂-SS31 in VEGF-induced cellular migration and permeability *in vitro* (**Fig. 4a-c**). The results confirmed the blocking ability of anti-VEGFR2 F(ab')₂ and anti-VEGFR2 F(ab')₂-SS31 in VEGF/VEGFR2 pathway. These findings provided additional evidence supporting the inhibitory effects of these antibodies on VEGF-induced functional phenomena.

Fig. 4a-c. Inhibition of cellular migration by anti-VEGFR2 F(ab')₂-SS31 *in vitro*.

High glucose-treated MRGEC or MPC5 cells were treated with anti-VEGFR2 F(ab')₂, SS31, or anti-VEGFR2 F(ab')₂-SS31 at a concentration of 0.4 μM for 12 or 24 hours.

(a) Light microscope images of the scratch edge in scratch assay and the migrated MRGECs in transwell migration assay after different treatments. Scale bar, 100 μm. (b and c) Quantitative assay of migration distance of scratch assay and the migrated cells counts of transwell migration assay in (a).

3. Response to comment: In most *in vivo* experiments, the anti-VEGFR2 (F(ab')₂) and

the anti-VEGFR2 (F(ab')₂)-SS31 showed equivalent results, although the author argues that the anti-VEGFR2 (F(ab')₂)-SS31 has greater efficacy. The text should be revised with this in mind.

Response: We apologize for the lack of rigor in our previous statements regarding the differences in therapeutic efficacy between anti-VEGFR2 F(ab')₂ and anti-VEGFR2 F(ab')₂-SS31. Both anti-VEGFR2 F(ab')₂ and anti-VEGFR2 F(ab')₂-SS31 showed significant therapeutic effects in DN mice, with several measured indices showing recovery to levels comparable to the sham group. In order to evaluate the difference in therapeutic efficiency between anti-VEGFR2 F(ab')₂ and anti-VEGFR2 F(ab')₂-SS31, we conducted a statistical analysis. As the reviewer mentioned, most *in vivo* experiments, including albuminuria and fibrosis tests, did not reveal significant differences in therapeutic efficacy between the two groups. However, when considering the impact of anti-oxidative stress factors, significant differences were observed (**Fig. 6a-d**). The results suggested that anti-VEGFR2 F(ab')₂-SS31 performs better than anti-VEGFR2 F(ab')₂ in terms of anti-oxidative stress. To address these concerns and improve precision, we have revised the relevant statements in Section “Treatment of DN” section (Line 269-307 of the revised manuscript).

Fig. 6a-d. Anti-VEGFR2 F(ab')₂-SS31 reduced oxidative stress in DN mice. (a) Immunohistochemical staining (brown) of nitrotyrosine, a marker of peroxidation. Scale bar = 100 μm. (b) Quantitative analysis of nitrotyrosine staining in (a). (c and d) SOD and MDA level changes in DN mice after different treatments. Data are expressed as the mean ± s.d. n = 6 in each group, n.s. no significant difference, * $p < 0.05$, ** $p < 0.01$, *** $p < 0.001$ as compared with DN group; n.s. no significant difference, # $p < 0.05$, ### $p < 0.001$ between groups as indicated.

4. Response to comment: In the results shown in Figure 6, the author has evaluated the levels of IL6 and TNF. There are other inflammatory cytokines that should have been tested for a larger panel, including interleukin 8, at least IL1beta.

Response: Thank you for your valuable advice regarding the comprehensive evaluation of the anti-inflammatory efficiency of anti-VEGFR2 F(ab')₂-SS31 through the supplementation of additional tests for inflammatory cytokines, specifically interleukin 8 (IL-8) and IL1beta (IL-1β). It is known that immune and inflammatory mechanisms play an important role in the development and progression of DN, which is recognized

as a chronic inflammatory disease [BMC Nephrol. 2020, 21, 308; Clin. Sci. (Lond). 2013, 124, 139-52]. Studies have reported that inflammatory cytokines, including IL-6, TNF- α , IL-1 β , and IL-8, are relevant to the development of DN [BMC Nephrol. 2020, 21, 308; J. Am. Soc. Nephrol. 2008, 19, 433–42]. Previous research has examined the levels of IL-6 and TNF- α (**Fig. R2a,b**), and in this study, we further investigated the levels of IL-1 β and IL-8 (**Fig. R2c,d**). The results indicated that the mice treated with anti-VEGFR2 F(ab')₂ and anti-VEGFR2 F(ab')₂-SS31 had lower expression levels of IL-6, TNF- α , IL-1 β , and IL-8, compared with nontreated or SS31 administrated DN mice and there was no significant difference compared to sham mice. The results of various inflammatory cytokine tests collectively suggested the strong anti-inflammatory capability of anti-VEGFR2 F(ab')₂ and anti-VEGFR2 F(ab')₂-SS31. Detailed revision can be found in Line 290-295 of the revised manuscript and Supplementary Fig. 6.

Fig. R2. Anti-VEGFR2 F(ab')₂-SS31 reduced inflammation in DN mice. Renal cytokine TNF- α , IL-6, IL-1 β , and IL-8 alternation in DN mice after different treatments. All data are expressed as the mean \pm s.d. n = 6 in each group, n.s. no significant difference, * $p < 0.05$, ** $p < 0.01$ as compared with DN group.

5. Response to comment: The author describes the presence of macrophages in general. However, did they test the relative M1/M2 macrophage ratio to possibly assess a precancerous state?

Response: Thank you for your kind and constructive advice about further investigating the macrophage M1/M2 polarization, which is reported to be associated with DN progress and a precancerous state. M1 macrophage promotes inflammation, and is linked to tissue damage and tumor cell killing [Nat. Rev. Drug Discov. 2022, 21, 799-820], whereas M2 macrophage suppresses immune reactions, participates in tissue repair and remodeling, and contributes to the proliferation and survival of cancer cells [Nat. Rev. Drug Discov. 2022, 21, 799-820, Annu. Rev. Pathol. 2020, 15, 123-147]. The ratio of M1/M2 macrophage is elevated in the early stage of DN, and decreases as DN progression [Front. Immunol. 2022, 13, 1015142]. The decreased ratio also creates a tumor-promoting microenvironment [Cell Metab. 2012, 15, 432-437]. In this study, we supplemented the immunofluorescence staining assay of M1 and M2 macrophages and calculated the M1/M2 ratio (**Supplementary Fig. 5**). The results showed weak signals of M1 and M2 macrophages and a relatively high M1/M2 ratio of approximate 5.7 in normal mice. In DN mice, the signals of M1 and M2 macrophages increased, and the M1/M2 macrophage ratio decreased to 2.5, indicating DN progression and a more tumor-promoting state. Mice treated with anti-VEGFR2 F(ab')₂ and anti-VEGFR2 F(ab')₂-SS31 had lower macrophage fluorescence and a higher ratio of M1/M2 macrophage (4.4 and 4.4 respectively) than DN mice that were not treated or were

administered SS31 (approximately 3.0), indicating that anti-VEGFR2 F(ab')₂ and anti-VEGFR2 F(ab')₂-SS31 effectively inhibited the DN progression and precancerous changing.

Supplementary Fig. 5. Anti-VEGFR2 F(ab')₂-SS31 reduced M1/M2 macrophage ratio in DN mice. (a) Immunofluorescence staining of CD86 (red) and CD206 (green) in DN kidneys after different treatments. M1 macrophage is stained with CD86 (red), and the M2 macrophage is stained with CD206 (green). Nuclei, staining with DAPI (blue). Scale bars: 50 μ m. (b) The mean ratio of fluorescence intensity of M1 and M2 macrophages in (a). All data are expressed as the mean \pm s.d. n = 6 in each group, n.s. no significant difference, * $p < 0.05$ as compared with the DN group.

6. Response to comment: The authors address fibrosis, which is very important considering that DN can cause dead-end fibrosis. Other fibrosis markers should have

been tested at least by qPCR (FAP, collagen, ...).

Response: Thank you for your kind advice. As you mentioned, renal fibrosis is a crucial metabolic change in DN, associated with the progression toward end-stage renal disease [J. Am. Soc. Nephrol. 2008, 19, 2282-2287]. The renal expression of fibrosis marker α -smooth muscle actin (α -SMA) has been tested (**Fig. 6i,j**). We further evaluated the expression of other typical fibrosis marker collagen I (the antibody of which was also available and convenient for us) directly by immunohistochemical staining (**Supplementary Fig. 7a**). The results revealed an upregulation of fibrosis factor collagen I, in the kidneys of DN mice, which subsequently decreased following treatment with anti-VEGFR2 F(ab')₂ and anti-VEGFR2 F(ab')₂-SS31. These results, in conjunction with the α -SMA assay, collectively confirm the anti-fibrotic efficacy of anti-VEGFR2 F(ab')₂ and anti-VEGFR2 F(ab')₂-SS31. We hope this response meets with your approval. Detailed revision can be found in Line 297-301 of the revised manuscript.

Supplementary Fig. 7a. Immunohistochemical staining (brown) of collagen I, marker of fibrosis. Scale bar = 50 μ m.

7. Response to comment: A very important parameter in this context is the fibrosis parameter. Among the cytokines involved in this parameter, TGF beta, CCN2/CTGF are leading cytokines that should be tested.

Response: Thank you for your specific suggestions regarding the fibrotic assay, which is helpful for strengthening the conclusion that anti-VEGFR2 F(ab')₂ and anti-VEGFR2 F(ab')₂-SS31 exhibit potent anti-fibrotic ability. We tested the leading cytokines involved in fibrosis, transforming growth factor β1 (TGF-β1) [Nature 2020, 587, 555-566] and connective tissue growth factor (CTGF) [J. Am Soc Nephrol. 2004, 15, 1430-1440] by ELISA (**Supplementary Fig. 7b,c**). The results indicated that the levels of TGF-β1 and CTGF were elevated in DN kidney and were significantly reduced following treatment with anti-VEGFR2 F(ab')₂ and anti-VEGFR2 F(ab')₂-SS31. These findings were coincident with the results of immunohistochemical staining assay of fibrotic markers, α-SMA, FAP, and collagen I.

Supplementary Fig. 7. (b and c) The expression changes of TGF-β1 and CTGF in DN mice after different treatments. All data are expressed as the mean ± s.d. n = 6 in each group, n.s. no significant difference, * $p < 0.05$, ** $p < 0.01$, *** $p < 0.001$ as compared with DN group.

8. Response to comment: There are several anti-angiogenics that have been approved in the clinic for the treatment of abnormal angiogenesis. How do the anti-VEGFR2 (F(ab')₂)-SS31 antibodies compare to the historical anti-VEGF Avastin/bevacizumab and/or multi-tyrosine kinase inhibitors, at least on cultured cells.

Response: Thank you for your kind and constructive advice. The anti-VEGF and

tyrosine kinase inhibitors you mentioned, including our anti-VEGFR2 F(ab')₂-SS31, all belong to the inhibitors targeting the VEGF/VEGFR2 pathway. Anti-VEGF acts on the upstream component VEGF, and anti-VEGFR2 and tyrosine kinase inhibitors act on the downstream component VEGFR2. Clinical products about anti-VEGF (e.g. Avastin/bevacizumab) and tyrosine kinase inhibitors (e.g. Gleevec/Imatinib) have been utilized for the treatment of abnormal angiogenesis in conditions such as cancer, degenerative eye diseases and other inflammation conditions. However, our choice of anti-VEGFR2 rather than anti-VEGF or a tyrosine kinase inhibitor for DN therapy was motivated by factors beyond the ability to block the VEGF/VEGFR2 pathway, specifically considering the role of the drug carrier. VEGFR2 is overexpressed on the surface of glomerular endothelial cells and podocytes of DN mice. Although tyrosine kinase inhibitors act on VEGFR2, they do not serve as suitable drug carriers due to a lack of specificity. By coupling the active cellular targeted ability of anti-VEGFR2 F(ab')₂ with the fragmentation-induced renal accumulation, we enhanced the distribution of the payload antioxidant peptide SS31 in targeted cells of kidneys. Moreover, these VEGF/VEGFR2 inhibitors exert their actions at different sites and have different effective concentrations, rendering direct comparisons less rigorous. We apologize for not supplementing our response with a comparative experiment. We hope this explanation meets with your approval.

Minor concerns:

1. Response to comment: The following sentence in the abstract is unclear: "We found

that the anti-VEGFR2 F(ab')₂ had higher accumulation in DN mouse kidneys and comparable VEGFR2 combination with the intact VEGFR2 antibody."

Response: Thank you for your careful review. We are sorry for the confusing statements, and have modified the sentence as "We found that the anti-VEGFR2 F(ab')₂ had a higher accumulation in DN mice kidneys than the intact VEGFR2 antibody, and simultaneously preserved the binding ability to VEGFR2", that is, F(ab')₂ fragmentation benefits the renal accumulation of anti-VEGFR2 antibody meanwhile does not destroy its affinity to VEGFR2. The revision can be found in Line 30-32 of the revised manuscript.

2. Response to comment: The last sentence of the introduction is too long.

Response: Thank you for your kind advice. We are sorry for the influent sentence. The last sentence of the introduction has been revised as "An antibody tended to be a targeted drug carrier owing to its specificity. We further prepared anti-VEGFR2 F(ab')₂-SS31, a first-in-class antibody-drug conjugate (ADC) that was developed by conjugating anti-VEGFR2 F(ab')₂ with the mitochondria-targeted antioxidant peptide D-Arg-dimethylTyr-Lys-Phe-NH₂ (SS31) [J. Am. Soc. Nephrol. 2011, 22, 1041-1052; Br. J. Pharmacol. 2014, 171, 2029-2050] to targeted treat DN. We investigated its efficiency in VEGFR2 blocking, anti-oxidative stress, and DN therapy *in vitro* and *in vivo*." (Line 66-71 of revised manuscript)

3. Response to comment: The "Conclusion/Discussion" part is very short and should be expanded to put the results in the general context of treating DN.

Response: Thank you for your kind suggestion regarding the expansion of the Discussion part. We have included the results of our DN treatment experiments in the discussion and analyzed the advantages and efficiency of anti-VEGFR2 F(ab')₂-SS31 in the treatment of DN. The following text has been added to the Discussion part of the revised manuscript.

ADCs have emerged as a potent strategy in cancer therapy, utilizing a target of interest as a delivery vehicle for toxic molecules [Curr. Opin. Immunol. 2016, 40, 14-23]. In this study, we aimed to validate the effectiveness of antibody fragments in the development of ADCs for DN therapy. Specifically, we present the development of a new anti-VEGFR2 ADC, called anti-VEGFR2 F(ab')₂-SS31, which incorporated the antioxidant peptide SS31 as a payload. This choice was based on the recognition that oxidative stress plays a crucial role in the progression of DN [Cancer Res. 2015, 75, 330-343; Antioxid. Redox Signal. 2016, 25, 657-684]. We demonstrated that fragmentation and SS31 conjugation did not weaken the affinity of anti-VEGFR2 to VEGFR2. Moreover, both anti-VEGFR2 F(ab')₂ and anti-VEGFR2 F(ab')₂-SS31 could efficiently distribute within the cytoplasm of MRGEC and MPC5 cultured in a high glucose medium, which mimicked the conditions of DN. Our investigation indicated that anti-VEGFR2 F(ab')₂ exerted inhibitory effects on the permeability and migration of VEGF-induced glomerular endothelial cells by suppressing the VEGF/VEGFR2 pathway. Moreover, anti-VEGFR2 F(ab')₂ exhibited potent therapeutic effects in DN mice models, including protection of glomerular structure, and reduction of pro-

inflammatory factors and fibrosis indices. Notably, anti-VEGFR2 F(ab')₂-SS31, in addition to the outstanding performance in the above aspects, confers the advantage in attenuating oxidative stress due to its SS31 conjugation.

Special thanks to you for your valuable comments.

Reviewer: 2

This study investigates the viability of utilizing monoclonal antibodies for renal therapy. In particular, it offers a new way to treat the disease by making anti-VEGFR2 F(ab')₂-SS31 gather together in the kidney. By removing the Fc fragment and conjugating SS31 to anti-VEGFR2, this is possible. Anti-VEGFR2 F(ab')₂-SS31 can exert inhibitory effects on cell migration, oxidative stress, and apoptosis through experimental validation conducted in vitro. Hence, I will now provide a more comprehensive explanation of my suggestions.

Response: Thank the reviewer for the positive comments and consideration “In particular, it offers a new way to treat the disease by making anti-VEGFR2 F(ab')₂-SS31 gather together in the kidney. By removing the Fc fragment and conjugating SS31 to anti-VEGFR2, this is possible”. We appreciate receiving your kind and valuable comments. We have studied the comments carefully and have made corrections which we hope to meet with approval.

All the responses here are in blue. All the changes have been highlighted by giving the text a yellow background in the revised manuscript.

1. Response to comment: It is recommended to provide additional details regarding the rationale behind the choice of the mitochondria-targeted antioxidant peptide SS31 in *in vitro* and *in vivo* experiments.

Response: Thank you for your kind and valuable suggestion regarding the rationale for selecting SS31, the mitochondria-targeted antioxidant peptide, as a drug payload. It has been reported that oxidative stress is a major determinant in DN progress [Antioxid. redox signal. 2016, 25, 657-684], with mitochondria as the main site of oxidative stress [Nat. Rev. Nephrol. 2018, 14, 291–312]. SS31, as a mitochondrial-targeted antioxidative peptide, has been shown to protect mitochondrial cristae from damage and promote oxidative phosphorylation by selectively binding with cardiolipin *via* electrostatic and hydrophobic interactions, thus exerting inhibitory effects on oxidative stress [J. Am. Soc. Nephrol. 2011, 22, 1041–1052; Br. J. Pharmacol. 2014, 171, 2029–2050]. Considering these properties, we chose SS31 as the payload to enhance the anti-oxidative stress effect of anti-VEGFR2 F(ab')₂ in DN therapy. We evaluated the effect of anti-oxidative stress *in vitro* and *in vivo* (**Fig. R3**). The results indicated that anti-VEGFR2 F(ab')₂ had few effects on anti-oxidative stress in high-glucose treated cells or DN mice. However, SS31 loaded anti-VEGFR2 F(ab')₂ effectively reduced ROS in the mitochondria of high-glucose treated cells, decreased the expression of oxidative stress marker nitrotyrosine and oxidative stress-related factors, SOD and MDA, in DN kidneys. These findings indicates that the conjugation of SS31 on anti-VEGFR2 F(ab')₂ successfully conferred anti-oxidative stress abilities to the therapeutic agent.

Fig. R3. Anti-oxidative stress by anti-VEGFR2 F(ab')₂-SS31 *in vitro* and *in vivo*. (a) Fluorescence images of mitochondrial ROS analysis with MitoSOX in H₂O₂-stimulated cells after different treatments. Scale bar, 20 μm. High glucose-treated MRGECs or MPC5 cells were treated with anti-VEGFR2 F(ab')₂, SS31, or anti-VEGFR2 F(ab')₂-SS31 at a concentration of 0.4 μM for 24 hours. (b) The MitoSOX mean fluorescence intensity of MRGECs and MPC5 cells in (a). (c) Immunohistochemical staining (brown) of nitrotyrosine, a marker of peroxidation. Scale bar = 100 μm. After initiation of STZ-induced DN, different agents (SS31, anti-VEGFR2 F(ab')₂, and anti-VEGFR2 F(ab')₂-

SS31) were administered intravenously at a dosage of 33 nmol/kg, once every three days for 5 weeks. After administration, kidneys were collected for evaluation. (d) Quantitative analysis of nitrotyrosine staining in (c). (e and f) SOD and MDA level changes in DN mice after different treatments. All data are expressed as the mean \pm s.d. $n = 3$ in each group, n.s. no significant difference, * $p < 0.05$, ** $p < 0.01$, *** $p < 0.001$ as compared with high glucose group or DN group; n.s. no significant difference, # $p < 0.05$, ## $p < 0.01$, ### $p < 0.001$ between groups as indicated.

2. Response to comment: If possible, it would be good to do more tests to show that anti-VEGFR2 F(ab')₂-SS31 has the potential to reduce proteinuria and blood creatinine levels in diabetic mice and even in humans.

Response: Thank you for your kind and valuable suggestions regarding the evaluation of anti-VEGFR2 F(ab')₂-SS31 in renal function recovery. Proteinuria, measured by urine albumin-to-creatinine ratio (UACR) test, is a common indicator of chronic kidney diseases, including DN. The presence of albumin in the urine indicates increased permeability across the glomerular filtration barriers, which is normalized to creatinine to control for variations in urine flow rate [J. Vis. Exp. 2018,136, 57764]. UACR has been widely used in clinical and preclinical studies [Am. J. Kidney Dis. 2014, 63, 713-35; J. Am. Soc. Nephrol. 2015, 26, 1889-904]. We have investigated the changing of UACR in DN mice after different treatments (**Fig. 5d**). The results demonstrated a significant decrease in UACR in DN mice treated with anti-VEGFR2 F(ab')₂-SS31, indicating the effective reduction of proteinuria. Serum creatinine, another potential

indicator of kidney disease progression, is less frequently employed in DN evaluation than proteinuria. Existing literature often focuses on proteinuria as the primary indicator of renal function recovery [Nat. Med. 2007, 13, 1349-1358; J. Am. Soc. Nephrol. 2020, 31, 1267-1281; J. Am. Soc. Nephrol. 2012, 23, 405-411; J. Am. Soc. Nephrol. 2007, 18, 2054-2061]. We think that the UACR results reflect the recovery of renal functions following treatment with anti-VEGFR2 F(ab')₂-SS31. We hope that this response meets with your approval.

Fig. 5d. Urinary albumin/creatinine ratios in different groups of mice after different treatments. Data are expressed as the mean \pm s.d. $n = 6$ in each group, n.s. no significant difference, ** $p < 0.01$, *** $p < 0.001$ as compared with DN group.

The antibody employed in our study is procured from Bio X Cell (Product name: *InVivoPlus* anti-mouse VEGFR-2, Clone: DC101, Isotype: Rat IgG1 κ , Immunogen: Mouse VEGFR-2-SEAPs soluble receptor). It is important to note that *InVivoPlus* anti-mouse VEGFR-2 is just for laboratory use and is not intended for any clinical, therapeutic, or diagnostic use in humans or animals. We appreciate your valuable suggestion regarding the clinical application of anti-VEGFR2 F(ab')₂-SS31. Our present study lays the groundwork for future clinical investigations into the efficacy of anti-VEGFR2 products for DN therapy. However, further advancements, such as

humanization, are necessary before considering clinical applications. We hope that this response meets with your approval.

3. Response to comment: There needs to be more talk about the important role of the anti-VEGFR2 F(ab')₂ fragment in renal therapy using SS31 technology in the conclusion section.

Response: Thank you for your kind and valuable comments regarding the significance of SS31 technology in renal therapy. In our study, we loaded SS31, the mitochondrial-targeted antioxidative peptide, in anti-VEGFR2 F(ab')₂, considering that oxidative stress is a major determinant in DN progress [Bri. J. Pharmacol. 2014, 171, 2029-2050; Antioxid. Redox Signal. 2016, 25, 657-684]. To assess its antioxidative stress capabilities, we conducted *in vitro* and *in vivo* experiments (**Fig. R3** mentioned in the response for Review 2 Comment 1). The results indicated that SS31 loaded anti-VEGFR2 F(ab')₂ effectively reduced ROS in mitochondria of high-glucose treated cells. Furthermore, it decreased the expression of nitrotyrosine, a marker of oxidative stress, as well as oxidative stress-related factors, SOD and MDA, in DN kidneys. These results indicated that SS31 loading effectively equipped anti-VEGFR2 F(ab')₂ with the anti-oxidative stress capabilities. We have supplemented the state in Discussion section (Line 332-334 of the revised manuscript).

4. Response to comment: There exist a few minor errors, such as inconsistencies in the style of references, which necessitate thorough scrutiny.

Response: We are sorry for this mistake and sincerely thank you for the careful review.

We have carefully checked the text and the style of references and corrected the errors throughout the manuscript.

5. Response to comment: The improvement of English language proficiency is necessary.

Response: Many thanks for the careful review and constructive suggestion. We do apologize for such errors. We have revised the manuscript and corrected the grammatical errors carefully.

Special thanks to you for your valuable comments.

REVIEWERS' COMMENTS

Reviewer #1 (Remarks to the Author):

The authors have responded to all the concerns expressed in the previous review

The revised version of the manuscript can then be accepted for publication.

Reviewer #2 (Remarks to the Author):

For the response manuscript, the authors and their team discussed in detail SS31 (mitochondria-targeted antioxidant peptide) as a drug payload, oxidative stress as a major determinant of DN progression, and mitochondria as the main site of oxidative stress. As a mitochondria-targeted antioxidant peptide, SS31 has been shown to selectively bind to cardiolipin through electrostatic and hydrophobic interactions, protect mitochondrial ridge from damage and promote oxidative phosphorylation, thereby producing inhibitory effects on oxidative stress.

Based on the above theoretical basis, the authors selected SS31 as a payload to enhance the anti-oxidative stress effect of anti-VEGFR2 F(ab')₂ in DN therapy. The authors evaluated the effects of anti-oxidative stress in vitro and in vivo and showed that anti-VEGFR2 F(ab')₂ had little effect on anti-oxidative effects. In addition, some clinical evidence was provided to support it.

At present, it seems that the author and his team have made a detailed discussion, which **may be considered for publication in your journal.**